WIND ENERGY SCIENCE DISCUSSIONS

# The Power Curve Working Group's Assessment of Wind Turbine Power Performance Prediction Methods

Joseph C. Y. Lee[1], Peter Stuart[2], Andrew Clifton[3], M. Jason Fields[1], Jordan Perr-Sauer[4], Lindy Williams[4], Lee Cameron[2], Taylor Geer[5], Paul Housley[6]

[1]National Wind Technology Center, National Renewable Energy Laboratory, Golden, Colorado 80401, USA
[2]Renewable Energy Systems, Kings Langley, Hertfordshire, England, UK
[3]Stuttgart Wind Energy, Institute of Aircraft Design and Manufacture, University of Stuttgart, Stuttgart, Germany
[4]Computational Science Center, National Renewable Energy Laboratory, Golden, Colorado 80401, USA
[5]DNV GL, Portland, Oregon, 97204, USA
[6]SSE plc, Glasgow, Scotland, UK

*Correspondence to*: Joseph C. Y. Lee (joseph.lee@nrel.gov)

**Abstract.** Wind turbine power production deviates from the reference power curve in real-world atmospheric conditions. Correctly predicting turbine power performance requires models to be validated for a wide range of wind turbines using inflow in different locations. The Share-3 exercise is the most recent intelligence-sharing exercise of the Power Curve Working Group, which aims to advance the modeling of turbine performance. The goal of the exercise is to search for modeling methods that reduce error and uncertainty in power prediction when wind shear and turbulence digress from design conditions. Herein, we analyze the data of 55 wind turbine power performance tests from 9 contributing organizations with statistical tests to quantify the skills of the prediction-correction methods. We assess the accuracy and precision of four proposed trial methods against the Baseline method, which uses the conventional definition of power curve with wind speed and air density at hub height. The trial methods reduce power-production prediction errors compared to the Baseline method at high wind speeds, which contribute heavily to power production; however, the trial methods fail to significantly reduce prediction uncertainty in most meteorological conditions. For the meteorological conditions when a wind turbine produces less than the power its reference power curve suggests, using power deviation matrices leads to more accurate power prediction. We also identify that for more than half of the submissions, the data set has a large influence on the effectiveness of a trial method. Overall, this work affirms the value of data-sharing efforts in advancing power-curve modeling and establishes the groundwork for future collaborations.

## 1 Introduction

Predicting the power output of a wind turbine for a given set of climatic conditions is a fundamental challenge in wind energy resource assessment. Current industry practices involve predicting power output using a power curve, which defines power production as a function of wind speed.

### 1.1 The challenge

Typically, a power curve is only strictly valid for a subset of all atmospheric conditions. For clarity, the Power Curve Working Group (PCWG, Sect. 2) refers to this subset of meteorological conditions as the "Inner Range." The corresponding "Outer Range" thus represents all other possible scenarios. The definitions are discussed in detail in Sect. 3.1.

A wind farm business case requires the power output to be predicted for the full range of meteorological conditions that the operational turbine will experience. Therefore, modeling approaches that accurately predict wind turbine power output in both Inner and Outer Range conditions are desirable to reduce the uncertainty associated with energy-yield predictions of future wind farms.

The wind energy industry performs power performance tests on wind turbines to test the site-specific power production of wind turbines by calculating the difference between the power predicted by the reference power curve and actual power production. However, these power performance tests and associated warranties are often limited to Inner Range conditions.

In reality, wind turbines operate in the Outer Range frequently, which sometimes leads to power-production deviations from the reference power curve. To quantitatively correct for such power deviations in different meteorological conditions, a power deviation matrix (PDM) is sometimes used (Fig. 1). Typically, when wind speed and turbulence intensity (TI, represents the deviations from the mean horizontal wind) are both low, the reference power curve overpredicts actual power production (bottom left quadrant of Fig. 1); when wind speed is low with high TI, the reference model would underpredict observed power (top left quadrant of Fig. 1); the observations are often incomplete for higher wind speeds (right half of Fig. 1). In practice, PDMs can be used to correct power prediction, some of which are illustrated in this study (Sect. 3.3 and Appendix A). Currently the industry lacks an objective criterion to evaluate correction methods of power deviation. Therefore, reaching an industry-wide consensus on the prediction method of wind turbine output in the Outer Range is necessary.

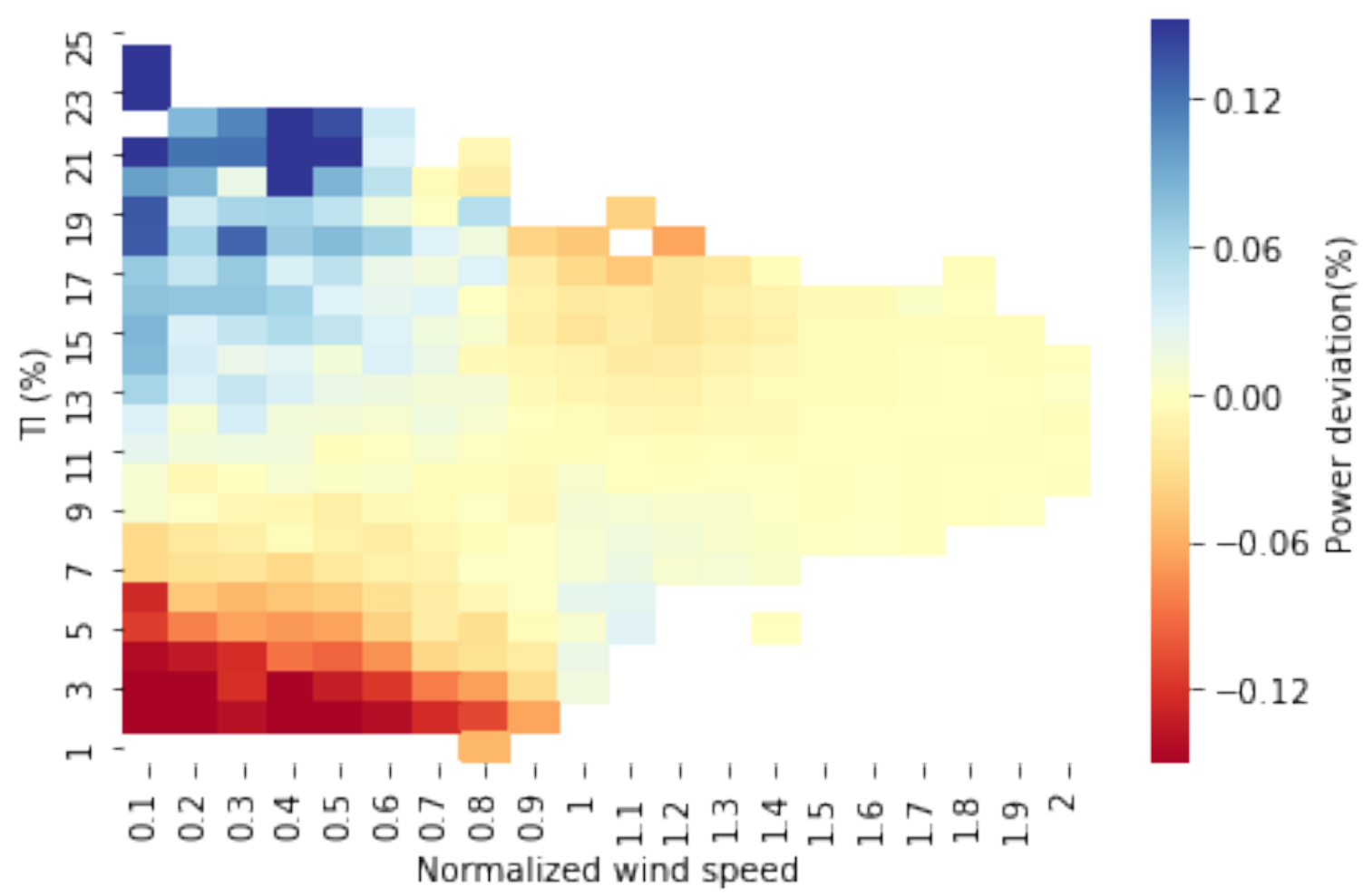

**Figure 1: A typical power deviation matrix (PDM) between normalized wind speed and turbulence intensity (TI). The predicted power subtracted from the observed power yields the power deviation in the Inner Range. A positive power deviation, seen in the blue region of low wind speeds and high TI, means larger observed power output than predicted power output, and vice versa for the red-colored cells. Zero normalized wind speed indicates the cut-in wind speed, and the normalized wind speed of one approximately equals to the rated wind speed. This particular PDM is derived and composited using 16 data sets supplied by a contributing member of the PCWG, and the data sets constitute part of the data submissions in this analysis.**

Additionally, the data that could be most useful for improving power-curve modeling are typically isolated within the industry, and their usage is stymied by intellectual property agreements. Thus, gathering this useful real-world data through intelligence-sharing initiatives can help improve our understanding of wind turbine performance in Outer Range conditions.

### 1.2 Candidate solutions

In 2005, an international standard on turbine power performance was published. The International Electrotechnical Commission (IEC) 61400-12-1 standard, Edition 1.0, 2005-12 (International Electrotechnical Commission, 2005) outlines the procedure of determining a power curve from measurements and executing a power performance test. Based on the 2005 standard, many power performance tests have been carried out and reported in the wind energy industry and academia. In 2017, the IEC updated the standard to Edition 2.0, 2017-03 (International Electrotechnical Commission, 2017), which includes standard methods for considering the influence of TI, wind shear, and wind veer in the power curve measurement. Because the IEC has not officially defined a standard power-curve prediction procedure for resource assessment, the industry often refers to the 61400-12-1 standards for power-curve modeling.

However, applying the standard in practice can be difficult. The 2017 standard describes theoretical prediction-correction methods for TI, wind shear (vertical change of wind speed), and wind veer (vertical change of wind direction). In reality, adoption of such analytical methods has not become the norm of the industry and a gap of implementation exists. Some of the cited methods only work for a limited set of power-production data sets and often are not applicable for wind resource assessment. Therefore, the industry still lacks a set of well-tested power-prediction correction methods that serves the purposes of both power performance testing and wind resource assessment. More importantly, not employing any corrections leads to increased scatter of production measurements of the power curve.

Moreover, the IEC standard considers wind speed as the primary variable, which can lead to poor power predictions, especially when wind turbines are waked (Ding, 2019). Research has since proven atmospheric variables other than wind speed and air density to be important in power modeling. Clifton et al. (2013) demonstrated that wind shear and TI impacted power performance with respect to the manufacturer's power curve in a clear and systematic way. They developed a machine-learning model, which includes shear and TI, and the model had about one-third the error in power prediction than using the method in the 2005 IEC standard (Clifton et al., 2013). Overall, accounting for various meteorological parameters, such as turbulence and atmospheric stability, enhances the skills in modeling power output and turbine loads (Bardal et al., 2015; Bardal and Sætran, 2017; Bulaevskaya et al., 2015; Hedevang, 2014; Sathe et al., 2013; Sumner and Masson, 2006; Wharton and Lundquist, 2012).

Introducing modern data with data-driven statistical methods to improve power modeling techniques has been the new direction of the wind energy industry. Past research demonstrates the benefit of using remote sensing and supervisory control and data acquisition (SCADA) data in power performance tests (Demurtas et al., 2017; Hofsäß et al., 2018; Mellinghoff, 2013; Rettenmeier et al., 2014; Sohoni et al., 2016; Wagner et al., 2013, 2014). Many experts use the PDM approach (Fig. 1) to observe any systematic bias in power curves and correct this in energy-yield models. PDMs can be generic, empirically derived, or turbine-model specific. The PDM approach is not documented in the IEC standard; nevertheless, the technique has been widely used in the industry. For instance, Whiting (2014) uses PDMs to validate wind turbine energy production. Recently, machine learning and neural networks that derive multidimensional power curve models have grown in popularity (Bessa et al., 2012; Jeon and Taylor, 2012; Lee et al., 2015; Optis and Perr-Sauer, 2019; Ouyang et al., 2017; Pandit and Infield, 2018; Pelletier et al., 2016).

It is clear that the industry intends to collectively advance our understanding of the power curve and model power performance with other variables beyond wind speed and air density. Hence, the PCWG was created to bridge academic research and industry practices.

## 2 The Power Curve Working Group

The mission of the PCWG is to bring together wind industry stakeholders to help identify, validate, and develop ways to improve the modeling of wind turbine performance in the real-world, complicated atmospheric conditions. The PCWG aims to decrease the perceived investment risk and uncertainty of investors by understanding the Outer Range scenarios when the actual turbine output deviates from the reference power curve. Ultimately, the PCWG intends to reduce the average cost of wind energy production through advancing the industry's understanding of the turbine power curve. Therefore, one of the key activities of the PCWG is the intelligence-sharing initiative, which allows the benchmarking of the effectiveness of various power-prediction methods.

Established in 2012, the PCWG (https://pcwg.org/) is led by industry experts and is open for any organization to join and contribute to. The PCWG includes wind farm developers, turbine manufacturers, consultants, and research institutions. The PCWG receives broad support from the wind energy industry and has a mandate to improve turbine-performance modeling; thus, the results shown in this study are highly impactful.

Since 2015, the PCWG has conducted several industrywide data-sharing studies (Table 1). In the Share-1 exercise, the PCWG encountered calculation problems that led to interpolation errors and erroneous outliers. In the following Share-1.1 initiative, the PCWG solved the problems and streamlined the participation process. In the Share-2 exercise, the PCWG found that a calculation error led to bias that overstates the skills of the two PDM methods. In the Share-3 exercise, the PCWG performed extensive tests to the analysis tool (Sect. 3.2) to minimize calculation errors. Therefore, Share-3 represents refined results submitted by PCWG collaborators that can be shared with confidence (Power Curve Working Group, 2018).

**Table 1: Timeline of PCWG's intelligence-sharing exercise.**

| Timeline | Share initiative | Number of data sets (Parentheses indicate number of remote-sensing data sets) | Correction methods | | | |
|---|---|---|---|---|---|---|
| | | | Hub-height methods | | | |
| | | | IEC TI | 2DPDM | 3DPDM | Augmented IEC TI |
| December 2015 | 1 | 50 (4) | ✓ | ✓ | | |
| September | 1.1 | 44 (11) | ✓ | ✓ | | |
| June 2017 | 2 | 47 (6) | ✓ | ✓ | ✓ | |
| December 2018 | 3 | 55 (3) | ✓ | ✓ | ✓ | ✓ |

✓ indicates method included in trial with at least 30 applicable data sets. The details of the correction methods are discussed in Sect. 3.3 and Appendix A.

This manuscript is the first peer-reviewed journal article that summarizes the intelligence-sharing efforts orchestrated by the PCWG, which publicly disseminates the findings and conclusions from the Share-3 exercise. Specifically, this study compares different correction methods of power prediction in various meteorological conditions. Building on this manuscript, the PCWG plans to deliver a tangible contribution on power curve advancement to the IEC-61400-15 group. Overall, the Share-3 initiative exhibits a collective effort of the wind energy industry to reduce bias and uncertainty of power prediction in the Outer Range. The results presented in this study are all from the Share-3 exercise, unless stated otherwise.

## 3 Evaluation of turbine performance prediction

### 3.1 Inner Range definitions

The PCWG categorizes wind conditions into the "Inner Range" and the "Outer Range" (Power Curve Working Group, 2013). In practice, the Inner Range represents a relatively narrow range of conditions that is predominant on typical wind turbine test sites. The Inner Range can thus be thought of as the range of conditions for which the turbine output can be expected to meet or exceed its reference power curve, in that the reference power curve is typically informed by performance under test-site conditions. Subsequently, in Inner Range conditions, a turbine is expected to generate 100% or greater of the annual energy production (AEP) using a reference power curve. The decomposition of all atmospheric conditions into the Inner Range and Outer Range is purely conceptual, and in principle the boundary of the Inner Range could be defined by any set and range of parameters.

Meanwhile, the turbine performance under Outer Range conditions is less well represented by the reference power curve defined in the Inner Range. In Outer Range conditions, a turbine would reach an AEP of less than 100% on average. The

Outer Range conditions include all possible scenarios that lead to deviations from expected production, and often result in lower power production than expected. Therefore, various correction methods have been proposed to improve the predictability of turbine performance in the Outer Range.

The PCWG differentiates Inner Range and Outer Range data based on the wind shear, calculated using the wind speeds between the lower blade tip and hub height, and the TI at hub height (Power Curve Working Group, 2018). For example, using the Inner Range definition A, a time period belongs to the Inner Range when the wind shear is between 0.05 and 0.25 and the TI is between 8% and 12% (Table 2). Herein, the definition of Inner Range and Outer Range only depends on turbulence and shear, and the PCWG activities exclude other variables in operational performance corrections, such as icing, blade degradation, and suboptimal performance. These definitions correspond to the conditions that would be expected in a power performance test carried out on a new turbine under ideal conditions. The PCWG uses the concept of Inner Range and Outer Range because this pragmatic approach is easy to define and simple to apply, and this method defines clear limits beyond which performance deviation can be expected.

**Table 2: Different Inner Range definitions.**

| Inner Range definition | Shear range | TI range |
|---|---|---|
| A | 0.05 – 0.25 | 8% – 12% |
| B | 0.05 – 0.25 | 5% – 9% |
| C | 0.1 – 0.3 | 10% – 14% |

### 3.2 The PCWG analysis tool

The PCWG member organizations have access to a large number of power performance test data sets and contractual power curve guarantees, which offers an excellent opportunity to verify the accuracy of trial methods. However, these data sets are commercially sensitive, and they cannot be shared directly. Therefore, PCWG members designed and developed an analysis tool to enable intelligence sharing, rather than requiring commercially sensitive data sets or contractual performance guarantees to be disclosed.

The analysis tool is open sourced via GitHub and written in the Python programming language. The tool is formally released and distributed in the form of an executable program to encourage wide adoption.

End users configure their own portfolio of power performance test data sets using a graphical user interface that enables the correction methods to be evaluated for each data set. Anonymized reports containing a summary of aggregated error metrics for each power performance data set are generated and can be sent to an independent aggregator (in this study, the National Renewable Energy Laboratory, NREL) for further analysis. This anonymous reporting and subsequent analysis by the PCWG aggregator allow PCWG members and the wind energy industry to form an objective view of the accuracy of trial methods, without requiring member organizations to share commercially sensitive data.

The workflow illustrated in Fig. 2 is common to all PCWG sharing exercises. Within the tool, the user performs the data set configuration and portfolio definition steps manually; all subsequent steps are performed automatically by the tool. As data set and portfolio configuration data are saved in a standardized format based on eXtensible Markup Language (XML), the user does not have to reconfigure data sets to contribute to subsequent PCWG share initiatives. The PCWG can thus test new correction methods without participants having to reconfigure data sets. An updated version of the analysis tool is released to users each time new methods are added. These new correction methods can then be evaluated in a further iteration of the sharing initiative. The correction methods tested in the Share-3 exercise are described in Sect. 3.3 and Appendix A.

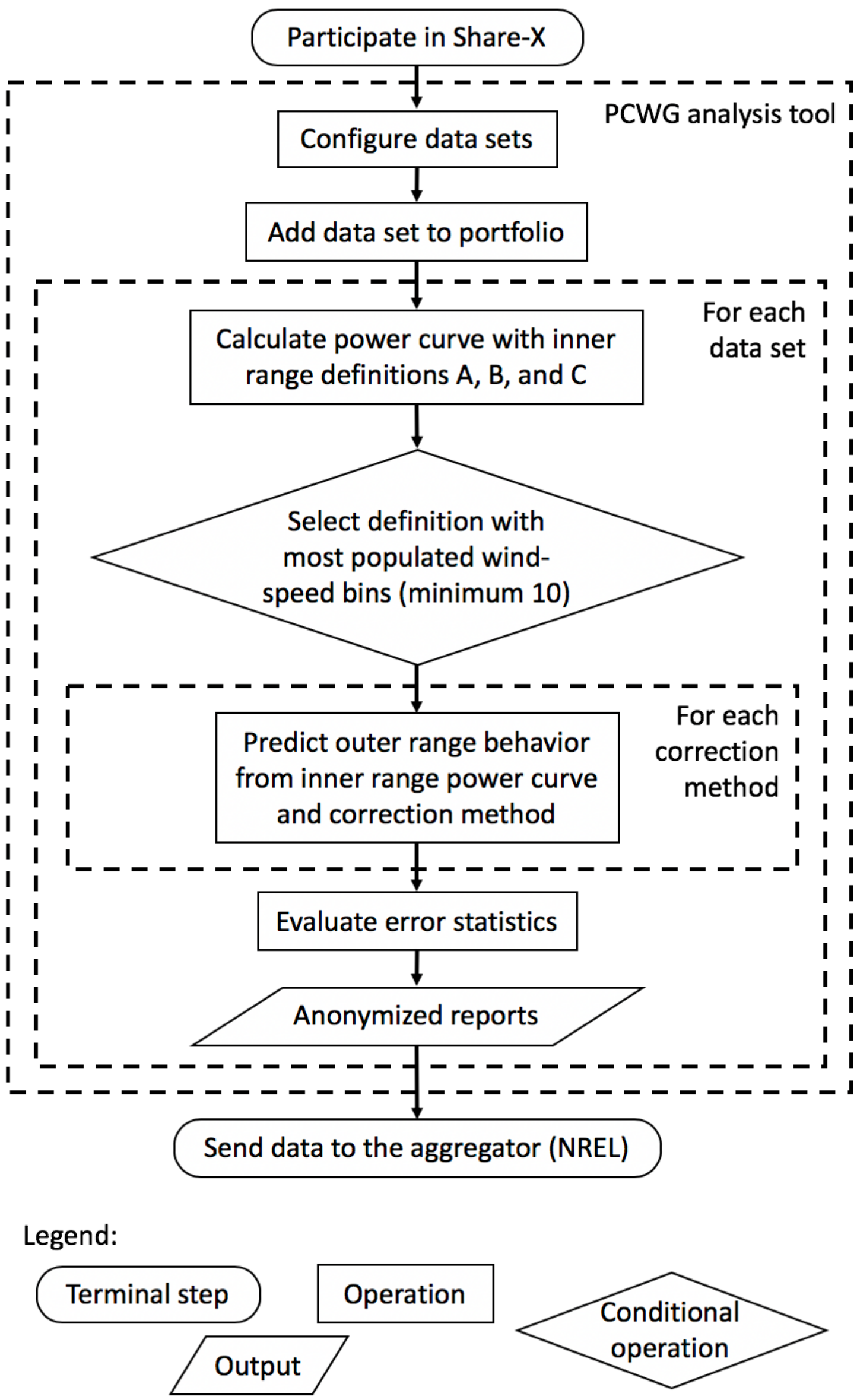

**Figure 2: Workflow of the Share-3 exercise.**

Each participant uses the analysis tool to produce human-readable results into one anonymized report for each data set in Microsoft Excel format. The error statistics (Sect. 3.4) of each correction method are aggregated in different categories (e.g., by normalized wind speed and time of day) in an Excel file. The participants then send the anonymized reports to the independent aggregator (NREL in the context of Share-3) for analysis.

For each data set, the PCWG analysis tool automatically selects an appropriate Inner Range definition (Table 1) depending on the 10-minute data counts in several atmospheric scenarios (Fig. 3). Next, the tool generates a power curve using an adequate amount of Inner Range data, which represents power production in a finite range of meteorological conditions. The resultant Inner Range power curve offers a basis to the power-prediction analysis, and this process resembles a measured power curve in reality based on a limited set of atmospheric cases. Then the tool applies the correction methods to predict turbine performance in the Outer Range with the Inner Range power curve. This extrapolation process requires a small but sufficient set of Inner Range data samples so as to predict the majority of data in the Outer Range. A poor Inner Range definition would classify all the data in the Inner Range and no data in the Outer Range.

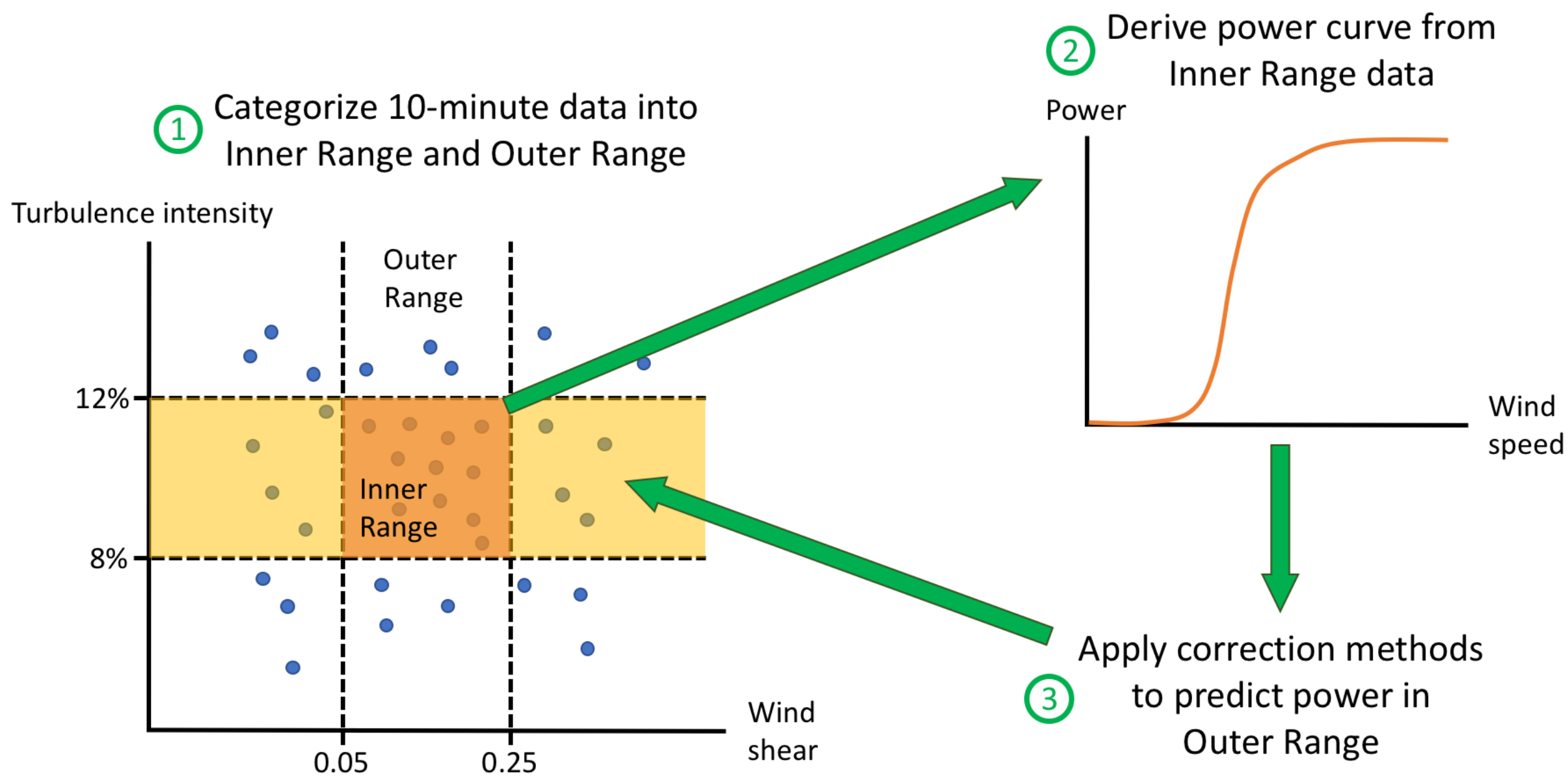

**Figure 3: How power curves are created and assessed in the Share-3 exercise. The orange and yellow boxes on the left represent the Inner Range, and the Inner Range TI with Outer Range wind shear, respectively.**

### 3.3 Correction methods

Several methods have been proposed in the IEC 61400-12-1 2017's standard (International Electrotechnical Commission, 2017) and elsewhere for postprocessing the data from a power performance test. These adjustments, often called correction methods, seek to account for the effect of changing atmospheric conditions on the wind turbine. One of the goals of the Share-3 exercise was to test the effectiveness of these methods (described in Table 3). Note that all five correction methods

use the density correction in the IEC 61400-12-1 2005 standard (International Electrotechnical Commission, 2005). Further details of the correction methods evaluated in this study can be found in Appendix A.

**Table 3: Abbreviation and key features of correction methods.**

| Correction method | Abbreviation | Trial method? | Key features |
|---|---|---|---|
| Baseline | — | No | Using interpolation based on the Inner Range power curve |
| Density and Turbulence | Den-Turb | Yes | Using the turbulence normalization method |
| Density and Two-Dimensional Power Deviation Matrix | Den-2DPDM | Yes | Using PDM to correct for the bins of normalized wind speed and TI |
| Density and Augmented Turbulence | Den-Augturb | Yes | Interpolating using regression across bins of normalized wind speed and TI |
| Density and Three-Dimensional Power Deviation Matrix | Den-3DPDM | Yes | Using PDM to correct for the bins of normalized wind speed, TI, and rotor wind speed ratio |

### 3.4 Error metrics and data categories

To contrast the accuracy of each power-prediction method, the Share-3 exercise uses two error metrics to evaluate each method, normalized mean error (NME) and normalized mean absolute error (NMAE) (Power Curve Working Group, 2018):

$$NME = \frac{\sum(P_{method}(t) - P_{actual}(t))}{\sum P_{actual}(t)} \quad (1)$$

$$NMAE = \frac{\sum|P_{method}(t) - P_{actual}(t)|}{\sum|P_{actual}(t)|} \quad (2)$$

where $P_{method}(t)$ is the modeled power calculated using any of the five methods mentioned in Sect. 3.3 and Appendix A for a given 10-minute period, and $P_{actual}(t)$ is the actual power production for a given 10-minute period. A perfect method would predict power matching the actual power production, and so NME would equal to 0 and NMAE would equal to 0. A positive NME means the correction method overpredicts power production. Generally, NME represents the average bias on power production of the correction method, which has a direct impact on P50, which is the average expected AEP (Clifton et al., 2016). Meanwhile, NMAE denotes the average cumulative error, which is applicable for short-term power-production forecasting and time series analysis, making NMAE a stricter metric than NME. For our purposes, we are interested in the long-term power prediction; hence, we only discuss the NME for the rest of this manuscript; NMAE is introduced here because the metric is also generated by the PCWG analysis tool (Sect. 3.2).

The PCWG analysis tool calculates NMEs (and NMAEs) by slicing the 10-minute data in several ways. For example, the overall NME yields a single value using all the available data for all atmospheric conditions. The Inner Range NME and

Outer Range NME include only the data from Inner Range and Outer Range, respectively. The tool also divides the data into different data categories based on inflow conditions:

- 15 normalized wind-speed bins, from 0 to 1.5, for all the data, the data in the Inner Range, and the data in the Outer Range;
- 4 wind-speed and turbulence-intensity (WS-TI) bins only for the Outer Range data, with four combinations of low wind speed (LWS), high wind speed (HWS), low turbulence intensity (LTI), and high turbulence intensity (HTI): LWS-LTI, LWS-HTI, HWS-LTI, and HWS-HTI. The threshold differentiating LWS and HWS is 0.5 normalized wind speed, and the TI threshold changes with the Inner Range definition of the data set (Table 2);
- wind direction;
- time of day;
- and calendar month.

In this study, we focus on contrasting the results from Inner and Outer Ranges, Outer Range normalized wind speeds, and WS-TI bins in the Outer Range to improve power predictions in the Outer Range.

Additionally, the bins in the Outer Range normalized wind speed and WS-TI data categories do not account for all the data in the Outer Range, thus we establish two new data bins for the residue samples. In reality, data with normalized wind speeds recorded above 1.5 exist, which exceeds the range between 0 and 1.5 in the setup of the PCWG analysis tool. Hence, those data below cut-in wind speed and beyond rated wind speeds are labelled as "Residual." Similarly, because we use wind shear and TI to classify Inner and Outer Ranges, the four basic WS-TI bins do not cover every data sample in the Outer Range, neglecting the data with Inner Range TI and Outer Range wind shear (ITI-OS) (yellow box in Fig. 3). Herein, we combine the analysis on the four WS-TI bins with the ITI-OS, and for each submission, the sum of the NMEs from these five data divisions is the Outer Range NME.

Moreover, we intend to examine the errors when the correction methods broadly impact the energy production in different meteorological conditions, especially at high wind speeds. Calculating NMEs using total energy integrated across all inflow conditions leads to larger NME variations in high wind speeds than in low wind speeds. Meanwhile, deriving NMEs from each confined data bin of a data category (for instance, the Inner Range, a bin, of the Inner-Outer Ranges, a category) results in larger NME variations in low wind speeds than in high wind speeds. This NME data per bin disproportionately skews the NMEs toward low wind speeds when a wind turbine does not generate power at its full capacity. Hence, we analyze the effects of the correction methods on total energy production throughout the whole power curve that spans between the cut-in and cut-out speeds.

To assess the impact on power production from each data bin of the categories, we also derive the energy fraction for every bin. From earlier, the PCWG analysis tool calculates the power-prediction errors based on both bin energy and total energy. Therefore, dividing the NME per total energy by the NME per bin energy yields the energy fraction a certain data bin represents in terms of total energy. For example, dividing the NME of the HWS-LTI bin per total energy by the NME of the HWS-LTI bin per its own bin energy returns the energy-production fraction of the HWS-LTI bin as a percentage across the WS-TI bins and the ITI-OS bin (Fig. 6a). Because wind turbines produce more power at higher wind speeds, the energy fraction

accounts for the shape of the power curve and weighs heavier toward HWS than LWS. Meanwhile, the data count of a data bin in a category only indicates the total number of 10-minute samples in that bin from the submission and does not account for the power production impact of that bin.

One of the goals of the Share-3 exercise is to identify the optimal methods in power prediction. To emphasize the trial method's improvement upon the Baseline method, we calculate the difference between the absolute value of the Baseline's NME and the absolute value of a trial method's NME. A negative difference means the method improves from the Baseline, and each method from each submission would result in different degrees of individual improvement.

### 3.5 Analysis methodologies

We perform several statistical tests to evaluate the trial methods' improvements from the Baseline method in different meteorological conditions, including the matched-pair t-test, the Levene's test, bootstrapping, and the Kolmogorov-Smirnov (K-S) test. The null hypothesis of the matched-pair t-test is that the trial method does not improve upon the Baseline in power prediction. When the null hypothesis is rejected, the improvement of the trial method upon the Baseline is statistically significant for that meteorological condition (Appendix B1). For the Levene's test, when the null hypothesis of a trial method is rejected, that method significantly decreases variance in prediction error from the Baseline (Appendix B2). This means the trial method reduces uncertainty in power prediction from the Baseline method in a specific inflow condition. Bootstrapping, which is resampling with replacement, generally is used to validate the results of the matched-pair t-test and the Levene's test. In this study, bootstrapped findings agree with the conclusions of the matched-pair t-test and the Levene's test; thus, the findings of the two statistical tests are representative (Appendix B3). The K-S test is to determine whether a sample distribution is Gaussian (Appendix B4). The details of the statistical tests are explained in Appendix B.

In this study, we cover and analyze all of the results, with and without statistical significance, from various statistical tests. For instance, even though some methods display improvement in predicting power from the Baseline method without statistical significance (the grey cells in Fig. 10b), we discuss the practical significance of how those methods compared with the Baseline in different atmospheric scenarios.

We also use filters to eliminate flawed data sets and increase the reliability of the statistical tests. We exclude erroneous submissions based on the nonzero Inner Range NMEs and the excess WS-TI 10-minute data counts (Appendix C1). We apply additional filters to achieve rigorous statistical inferences by removing data sets with substantial improvements from the Baseline (Appendix C2) and by implementing the Bonferroni correction to reduce alpha in statistical tests (Appendix C3). The filtering techniques we carried out are described in Appendix C.

## 4 Results and discussion

### 4.1 Metadata summary

We received 55 submissions from 9 organizations from the Share-3 exercise. About half of the submissions use turbines with rotor diameters between 86 m and 97 m, hub heights between 77 m and 88 m, and specific power between 299 W $m^{-2}$ and 347 W $m^{-2}$ (Fig. 4a, b, and c). Specific power is defined as the rated power divided by the swept area of the rotor. Almost half of the submissions are dated from 2015 (Fig. 4d). Overall, most of the turbines tested in the submissions, which represent the fleet installed, use modern control systems, so this is a pertinent study. Around half the participants chose to share the countries where their turbines were installed. Therefore, we know that this study includes data from Germany, Mexico, South Africa, Spain, the United Kingdom, and the United States. Hence, this analysis accounts for meteorological conditions at locations across the world.

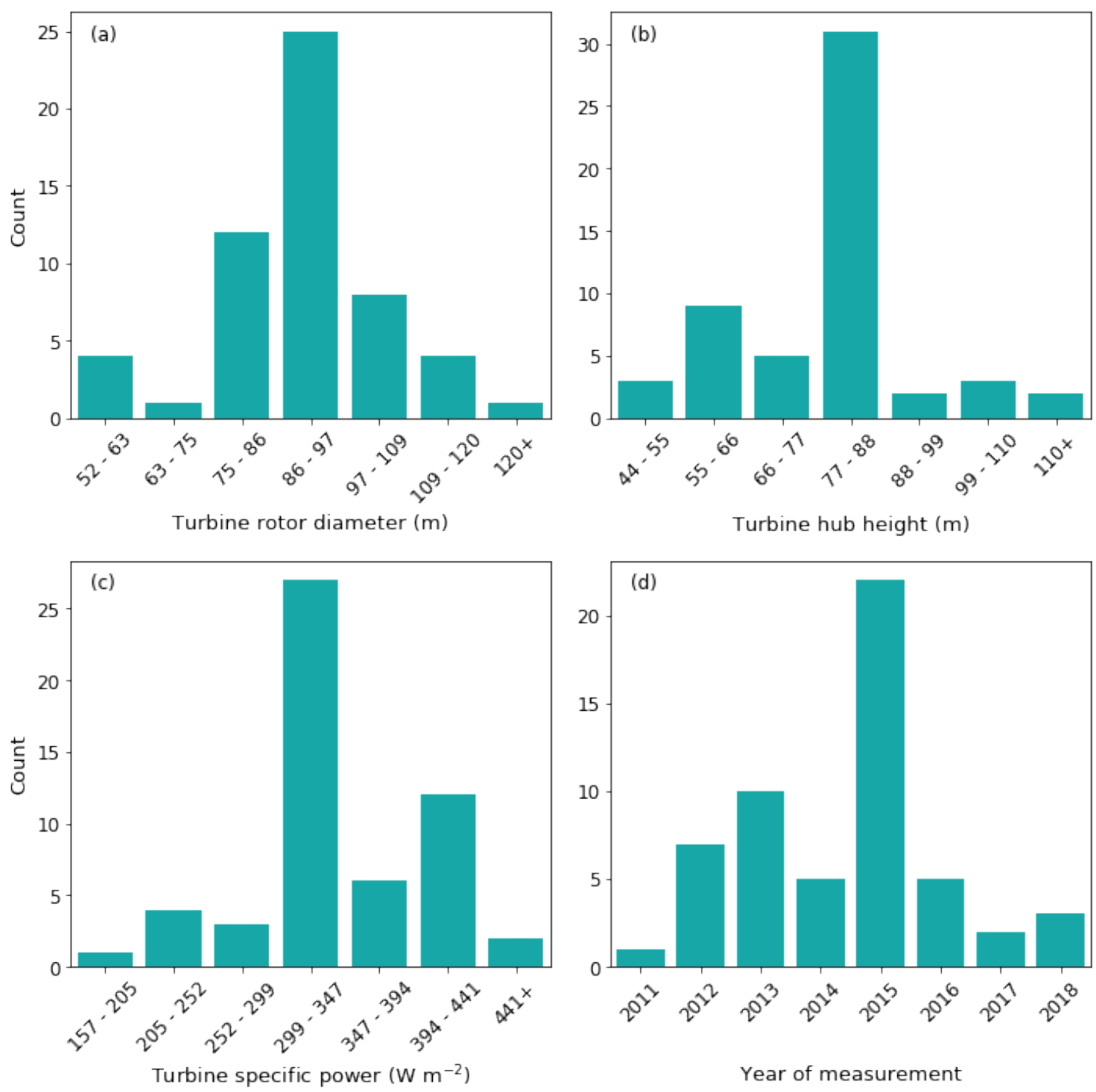

**Figure 4: The 55 submissions included turbines with rotor diameters from 50 m to 154 m (a), hub heights from 44 m to 143 m (b), and specific power from 157 W $m^{-2}$ to 583 W $m^{-2}$ (c). The tests were performed between 2011 and 2018 (d). These histograms display results without any filtering discussed in Appendix C.**

In some scenarios, the 10-minute data counts of the submissions have notable implications. For instance, the number of the 10-minute data sample in the Outer Range is larger than that in the Inner Range for all of the submissions (Fig. 5a). In three submissions, the sample size of the 10-minute Outer Range data is more than seven times than that of the Inner Range (Fig. 5b). Note that the NME filter (Appendix C1 and Fig. C1) is applied to remove erroneous submissions from all the results presented for rest of the manuscript.

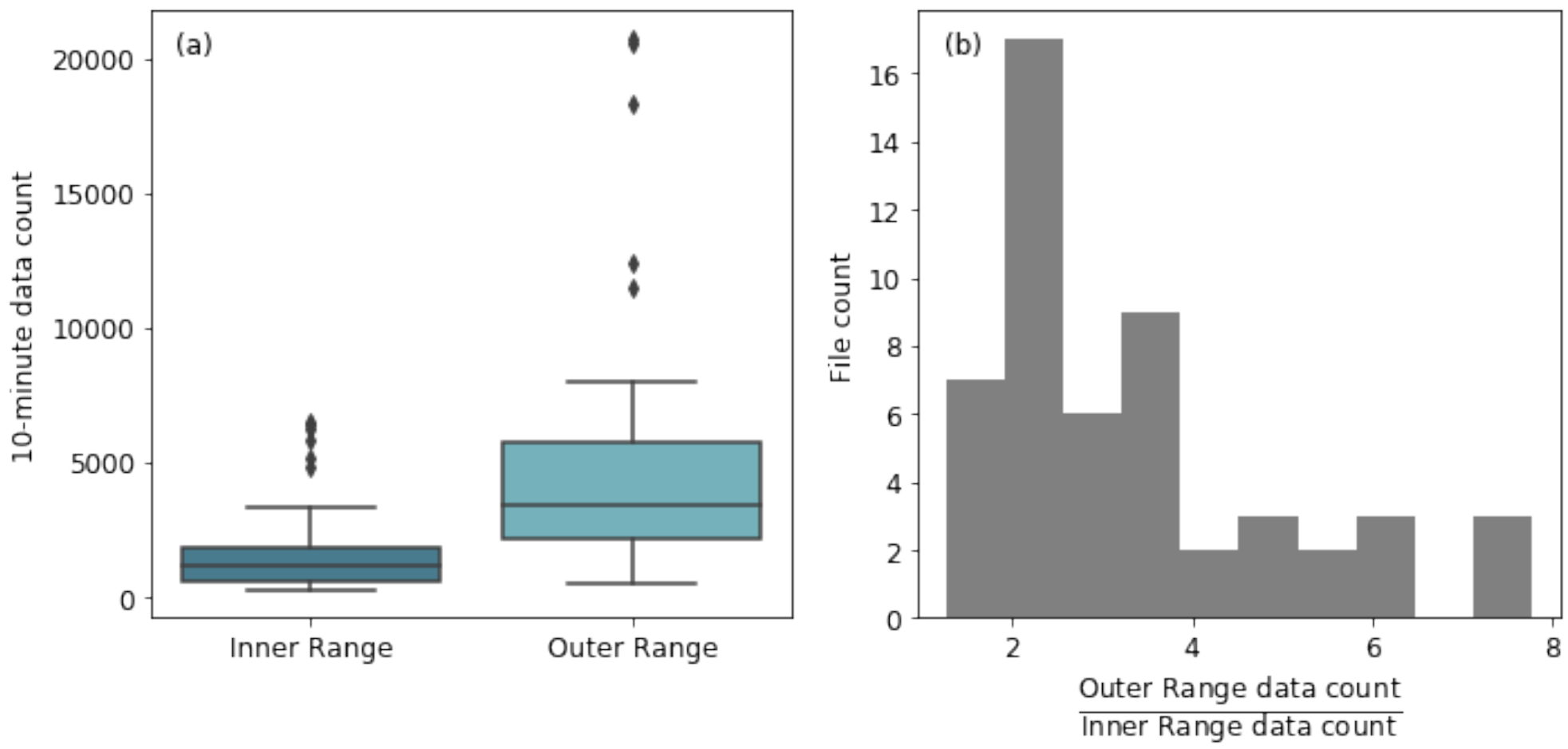

**Figure 5: (a) Boxplots of the 10-minute data count from the 52 NME-filtered submissions in Inner Range (dark blue) and Outer Range (light blue). The boxplot displays lower whisker (lower quartile minus 1.5 times of the interquartile range, which is the difference between the upper quartile and lower quartile), the lower quartile, the median, the upper quartile, the upper whisker (upper quartile plus 1.5 times of the interquartile range), and outliers as black diamonds; (b) histogram of the ratio between the Outer Range data count and the Inner Range data count from the 52 NME-filtered submissions.**

The majority of the data samples are classified as Outer Range, which meets our expectations. The PCWG analysis tool is able to classify a sufficient amount of Inner Range data to derive an Inner Range power curve for every data set, and the large amount of Outer Range data samples establishes a foundation to test the accuracy of the extrapolation process in power-production prediction (Fig. 3). After all, the large ratio between Outer Range and Inner Range data demonstrates that the Share-3 exercise is robust because of the large amount of Outer Range data for testing. Furthermore, the Inner Range data count does not correlate with the Outer Range NME regardless of the correction methods (not shown).

### 4.2 Energy fractions and NME distributions

The distributions of the 10-minute data counts are comparable in the four WS-TI bins in the Outer Range, whereas for many data sets, the HWS conditions contribute substantially more to turbine energy production than LWS scenarios (Fig. 6a). This feature fits our expectation because of the cubic relationship between wind speed and power. The Outer Range data also account for at least half of the energy production for most of the submissions (Fig. 6b), which is reasonable given the Outer Range data counts outweigh those of the Inner Range (Fig. 5). Overall, HWS conditions in the Outer Range, regardless of the TI, particularly deserve our attention on power-prediction correction.

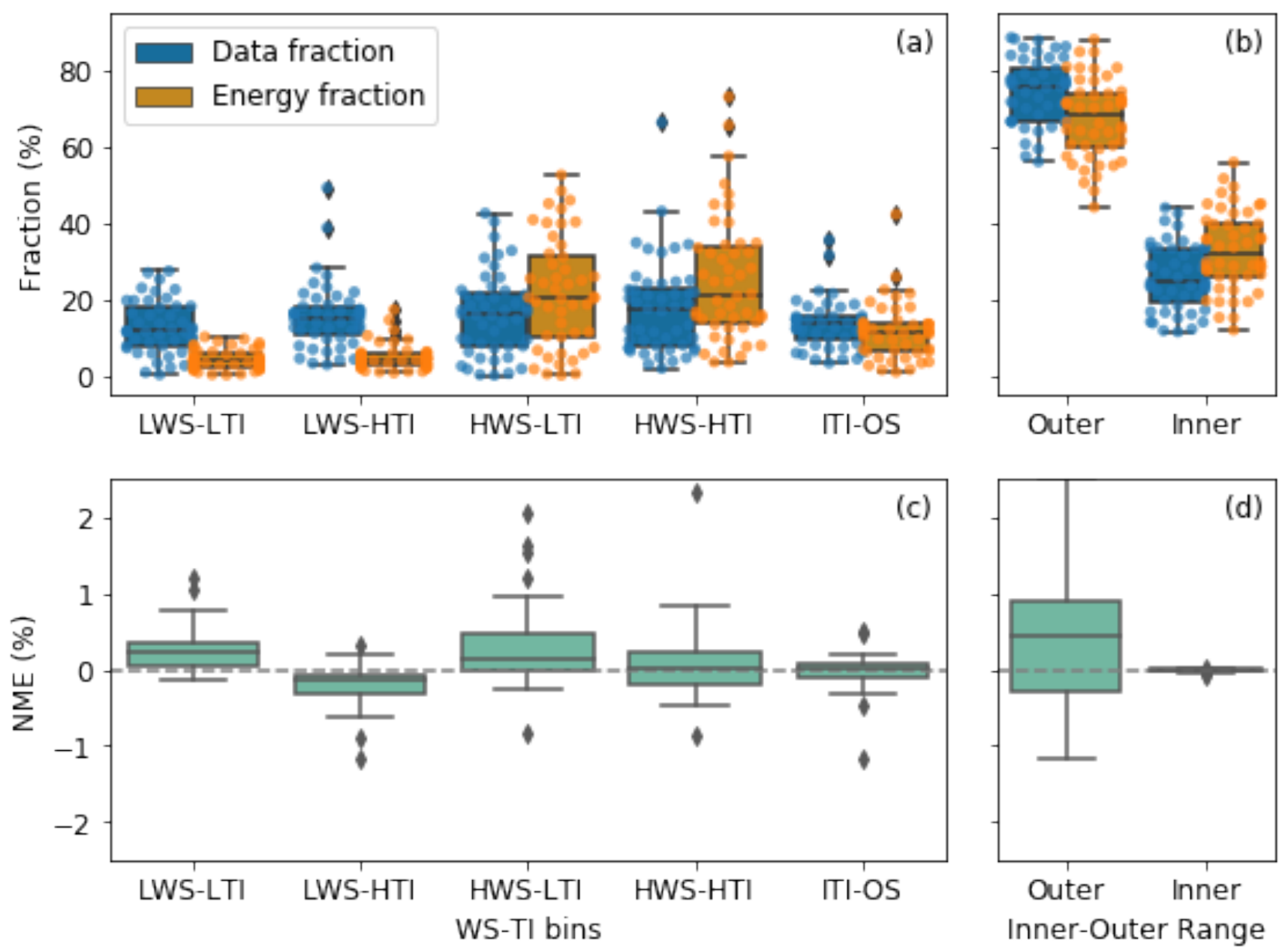

**Figure 6: (a) Boxplot of data fraction (blue) and energy fraction (orange) in percentage of the submissions across the four WS-TI bins and the ITI-OS bin for the Baseline method. Each colored dot in a bin represents a submission; (b) similar to (a), but for Inner and Outer Ranges. The dots represent the outliers, as in Fig. 5. For each submission, the sum of the fractions of the four WS-TI bins and the ITI-OS bin in (a) equals to the fraction of Outer Range in (b); (c) boxplot of the Baseline's NME in percentage across the same set of WS-TI and ITI-OS bins for the Baseline as in (a). The grey dashed line marks the zero NME, which theoretically is a perfect correction method. The range of NME shown is smaller than the observed, which provides a clearer perspective to contrast different WS-TI bins; (d) similar to (c), but for Inner and Outer Ranges. Similarly, for each submission, the sum of the NMEs in (a) equals to the NME of Outer Range.**

The data and energy fractions remain the same across correction methods for each submission, and the distribution shapes of NMEs across correction methods are analogous; thus, we use the Baseline's data fractions, energy fractions, and NMEs as an example in Fig. 6. In this manuscript, only 48 submissions are included in the WS-TI analysis after we apply the filtering techniques mentioned in Appendix C1. Moreover, not all of the submissions record 10-minute data in all the bins of different atmospheric categories (including the WS-TI category), because some specific wind conditions did not take place during the measurement periods.

Echoing the WS-TI energy fractions, the data with normalized wind speeds above 0.6 demonstrate extensive impact on energy production, even though they have smaller representation in the 10-minute data than those with lower wind speeds (Fig. 7a). The disproportionate energy-production contribution in the Outer Range is prominent especially for the samples with normalized wind speeds between 0.9 and 1.2. As mentioned, the analysis tool uses normalized wind speed of approximately 0.5 to differentiate LWS and HWS data. Therefore, we favor the correction methods that are effective at higher normalized wind speeds.

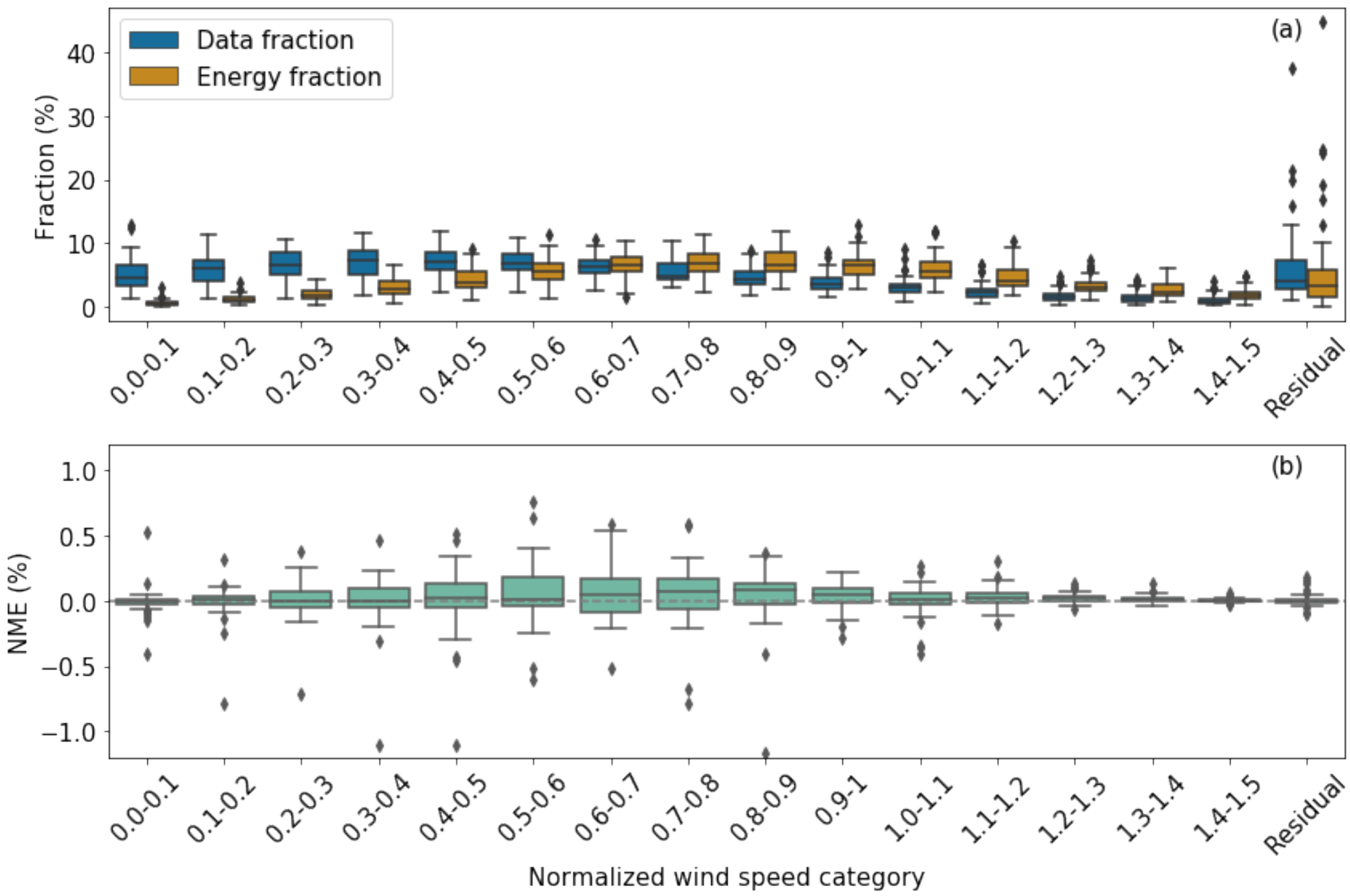

**Figure 7: Similar to Fig. 6, but for normalized wind speed bins in the Outer Range, without the colored dots in Fig. 6 (a). For clarity, the submissions are aggregated as boxplots and not displayed as dots in (a). For each submission, the sums of the data fractions, energy fractions, and NMEs from all the bins, including Residual, equal to those of Outer Range (in Fig. 6).**

Across correction methods, the average NMEs vary with different WS-TI and Inner-Outer Range bins, except for ITI-OS (Fig. 8a). When the TI is in the Inner Range and the wind shear is in the Outer Range, all the correction methods result in power underestimation. For the HWS bins of the Baseline method, the median NMEs tend to be weakly positive (Fig. 6c and 7b), which means the correction methods overestimate the real power production in the linear part of the power curve. However, the Baseline also yields the lowest error on average for HWS-HTI condition. Meanwhile, Den-2DPDM, Den-Augturb, and Den-3DPDM yield relatively low errors in the three bins with HWS and HTI, which impact energy production extensively.

eawe european academy of wind energy
Open Access WIND ENERGY SCIENCE DISCUSSIONS

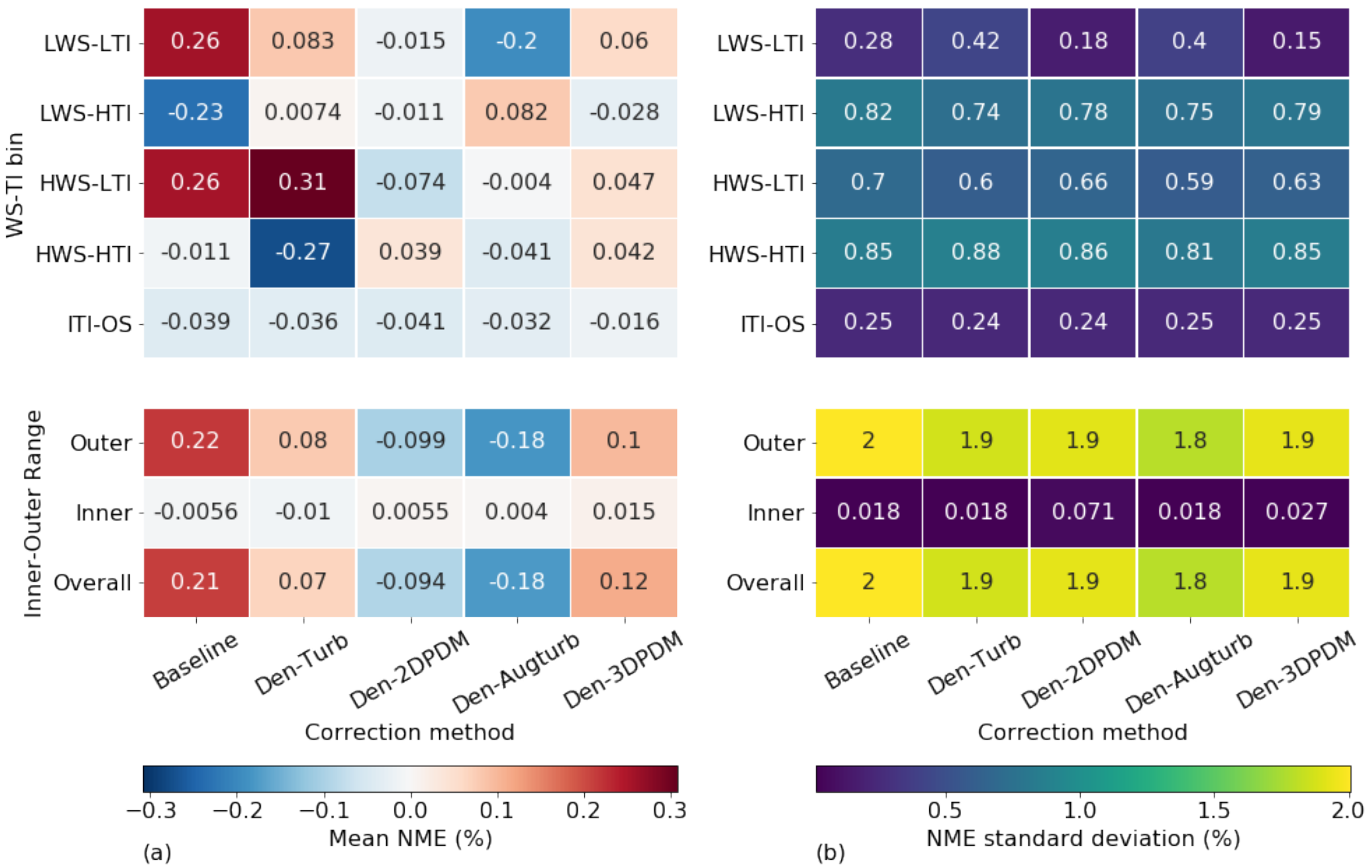

**Figure 8: (a) Heat map of mean NME based on 48 submissions across the four WS-TI bins and the ITI-OS bin in the Outer Range, along with the Inner and Outer Ranges, for 5 correction methods, where blue is negative and red is positive; (b) heat map of NME standard deviation using 48 submissions across WS-TI and Inner-Outer Range bins and correction methods. The annotated number in each cell represents the mean NME or the NME standard deviation of a specific set of data bin and correction method.**

Overall, the large average Outer Range NME of the Baseline method indicates sizable room for improvement in the power-prediction correction methodology. Additionally, the postfiltering Inner Range NMEs are close to zero (Fig. 6d and 8a), which aligns with the Inner Range definitions (Sect. 3.1 and Appendix C1).

Variations of NMEs among submissions are the smallest in the LWS-LTI bin and are substantially higher in the HWS and HTI bins (Fig. 8b). Moreover, the standard deviation of the Outer Range NMEs are about an order of magnitude larger than the NME averages. The large variation of the correction methods' errors demonstrates that the adjustments of power prediction are imprecise and remain uncertain.

We discuss the NME distributions of each correction method individually thus far. In the following section, we contrast the improvements of the four trial correction methods upon the Baseline method and perform statistical tests.

eawe european academy of wind energy
Open Access WIND ENERGY SCIENCE DISCUSSIONS

### 4.3 Improvements upon the Baseline method

#### 4.3.1 Impact of data sets

The performance of a trial correction method sometimes broadly depends on the input data set. The effectiveness of a trial method compared to the Baseline method varies greatly within a data set as well as among data sets (Fig. 9a). The effects of changing correction methods are limited on some data sets. Particularly, 30 of the submissions report less than 0.5% in the statistical range of the absolute-NME differences between the Baseline and the trial methods (Fig. 9b). The trial methods tend to yield similar results for a majority of the data sets (Fig. 9c). This means for more than half of the submissions, the choice of the trial methods has little impact on the resultant improvement or worsening against the Baseline method. For those cases, the data set itself dictates whether a trial method works or not: when a trial method is effective and becomes better than the Baseline, the other three trial methods would also yield comparable prediction corrections, and a similar phenomenon exists for the submissions with mixed and worsening signals.

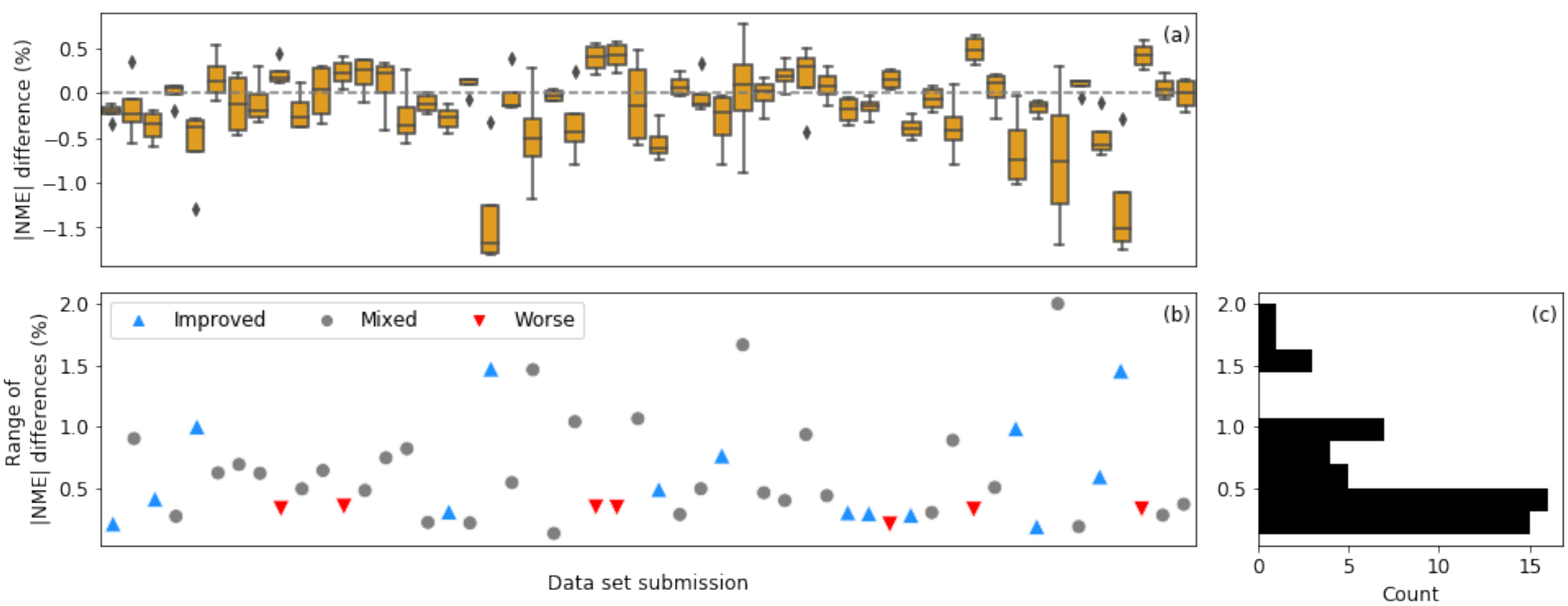

**Figure 9: (a) Boxplot of differences of absolute NMEs in the Outer Range between the Baseline method and each of the trial methods for the 52 data set submissions. Each box represents one submission, which has four data points of the Baseline-trial-method comparison; (b) scatterplot of the statistical ranges of the absolute-NME differences for the submissions. Each data point depicts the difference between the maximum and the minimum of the absolute-NME differences of each submission, corresponding to the boxes in (a). Submissions with all four negative absolute-NME differences against the Baseline in (a), i.e., improvements from the Baseline across trial correction methods, are shown as "Improved," those with four positive values in (a), i.e., deteriorations from the Baseline method regardless of trial method chosen, are shown as "Worse," and those submissions with no clear improvement or worsening are shown as "Mixed"; (c) histogram of the ranges in (b), using the same range on the vertical axis.**

Turbine characteristics are generally irrelevant to the performance of trial methods. Across trial methods, the magnitude of improvements upon the Baseline method does not correlate to any turbine characteristics (not shown), including turbine hub height and turbine specific power. The 14 submissions in which the four trial methods all improve from the Baseline (blue up-pointing triangles in Fig. 9) include a variety of turbine models. Meanwhile, the seven submissions that the trial methods strictly perform worse than the Baseline (red down-pointing triangles in Fig. 9) use turbines with rotor diameters

eawe european academy of wind energy
Open Access WIND ENERGY SCIENCE DISCUSSIONS

between 77 m and 100 m. Because of the lack of high-quality metadata, we cannot explain why some data sets record only improvements against the Baseline while some report the opposite.

### 4.3.2 Outer Range WS-TI and binned wind speed analysis

In the Outer Range, the four trial correction methods demonstrate stronger improvements against the Baseline method in LWS conditions than in HWS cases. More than 60% of the submissions report prediction error reduction by switching to a trial method from the Baseline for LWS cases (Fig. 10a), whereas this quantity is smaller for HWS and ITI-OS scenarios. For the LWS-HTI condition, the improvements are statistically significant across trial methods (Fig. 10b). Only Den-2DPDM and Den-3DPDM significantly reduce prediction-error uncertainty for LWS-LTI condition by lowering the NME variances from the Baseline's. The trial methods are more skillful than the Baseline for LWS.

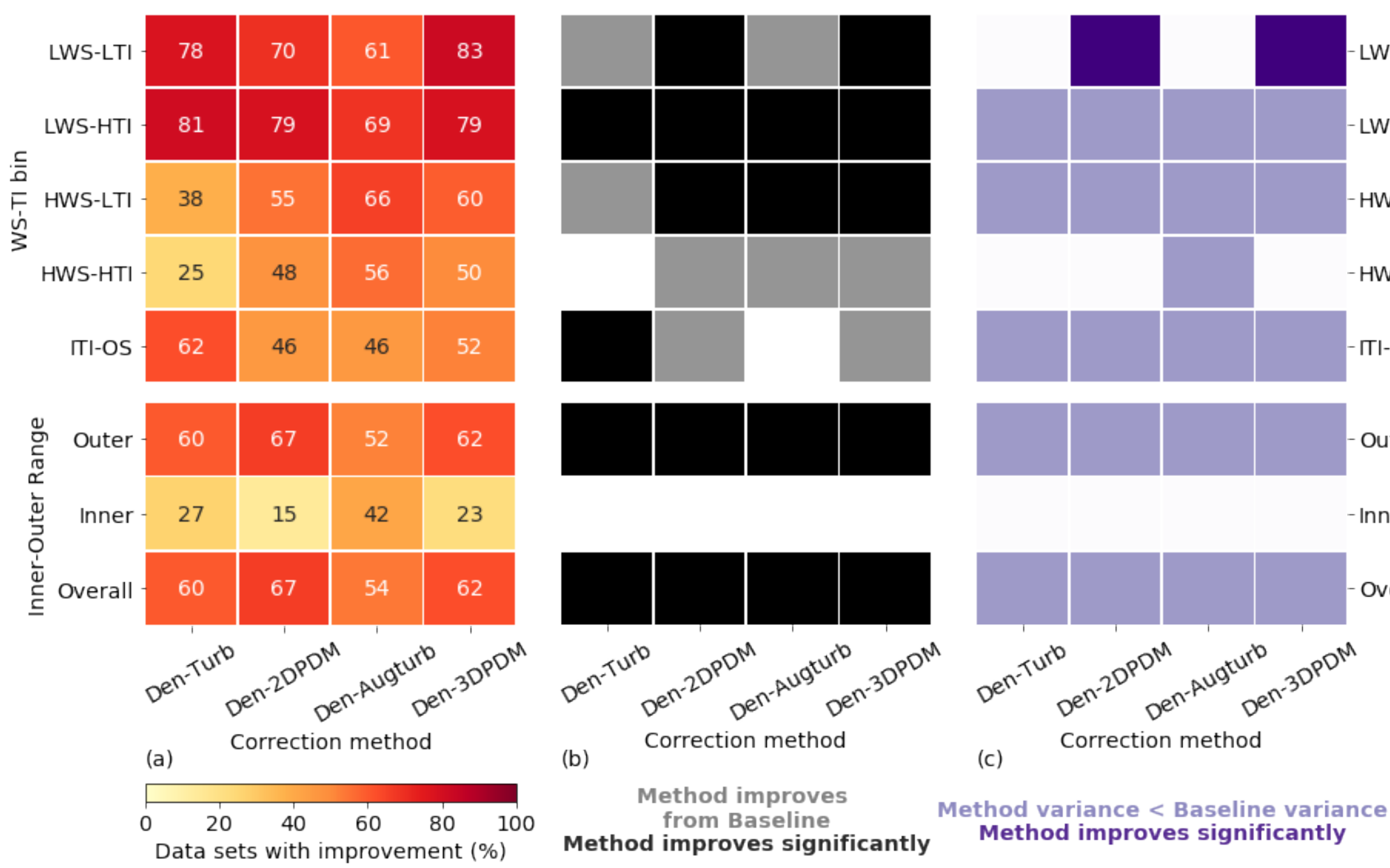

**Figure 10: (a) Heat map of the four trial methods' improvement fractions upon the Baseline method for the four WS-TI bins and the ITI-OS bin in the Outer Range and the Inner and Outer Ranges, calculated by combining the differences of absolute NMEs from individual submissions. The numbers in each cell annotate the individual improvement percentage; (b) heat map illustrating whether a trial method yields smaller absolute NME than the Baseline on average in each data bin (grey) or not (white), and whether the result is statistically significant after performing the one-sided matched-pair t-test with an alpha of 0.05 (black); (c) heat map representing whether the NME variance of a trial method is smaller than the NME variance of the Baseline method in each data bin (light purple) or not (white), and whether the result is statistically significant after performing the Levene's test with an alpha of 0.05 (dark purple).**

However, HWS scenarios in the Outer Range influence energy production more than other inflow conditions (Fig. 6c and 7a), and only Den-2DPDM, Den-Augturb, and Den-3DPDM perform significantly better than the Baseline in the HWS-

LTI condition (Fig. 10b). After making the t-test and Levene's test more rigorous by removing outliers and reducing alpha (Sect. 3.5, Appendix C2 and C3), the trial methods are barely better than the Baseline in HWS cases (Fig. D1). Hence, modifying the trial correction methods to effectively correct for prediction errors in HWS conditions will be a key objective for the next intelligence-sharing exercise.

In the Outer Range, the trial correction methods display stronger average performance improvements and larger uncertainty reduction from the Baseline than in the Inner Range. At least half of the submissions benefit from choosing a trial method to predict Outer Range power production over the Baseline (Fig. 10a). All of the trial methods statistically significantly reduce average NME from the Baseline in the Outer Range (Fig. 10b). All of the trial methods also reduce power-prediction uncertainty from the Baseline but are not statistically significant (Fig. 10c). After applying strict filters for the statistical tests (Sect. 3.5, Appendix C2 and C3), none of the improvements or uncertainty reduction remain statistically significant (Fig. D1). Additionally, the trial methods are far less useful in the Inner Range, yet the Outer Range constitutes over half of the data samples and energy production, so we primarily consider the methods' performance in the Outer Range.

Summarizing all meteorological conditions, all of the trial correction methods improve upon the Baseline method by yielding smaller overall errors. Each trial method results in overall NMEs closer to zero than the Baseline, and more than half of the submissions gain skills in power prediction by choosing a trial method over the Baseline (last row in Fig. 10). Although all the methods reduce the overall power-prediction uncertainty from the Baseline, the reductions in error variance are statistically insignificant. In general, applying a trial correction method leads to better power-production prediction on average, yet the precision of the prediction does not drastically improve. The trial methods have room for improvement in modeling power curves.

In the Outer Range, the four trial methods perform better than the Baseline method for nearly all of the wind speeds within a power curve. Given that normalized wind speeds above 0.6 are critical for energy production (Fig. 7a), all trial methods yield significantly better predictions for over half of the submission than the Baseline for normalized wind speeds between 0.6 and 0.8 in the Outer Range (Fig. 11a and b). Even though the trial methods are able to reduce prediction uncertainty across most wind speeds, the reductions are statistically insignificant (Fig. 11c).

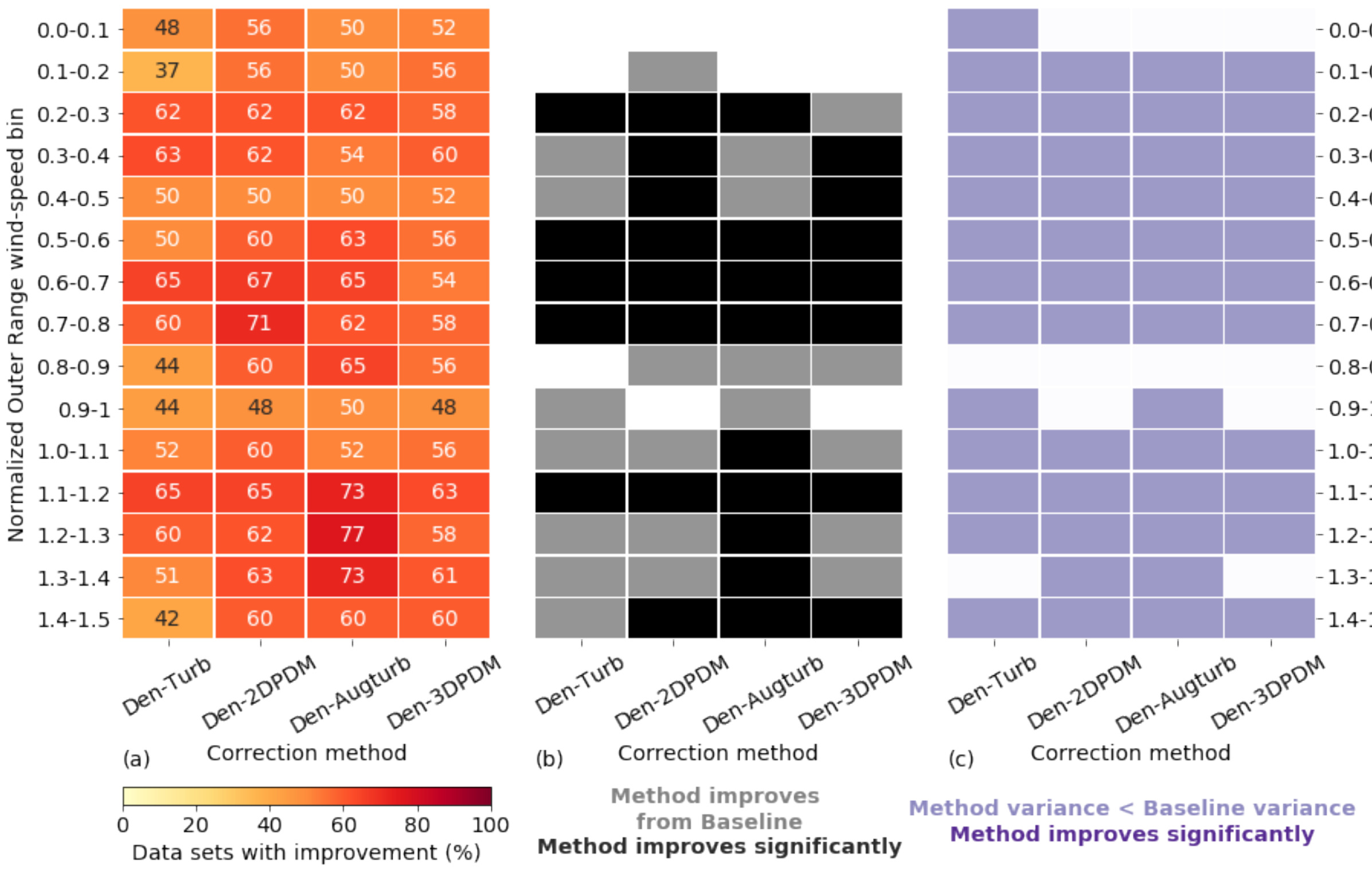

**Figure 11: As in Fig. 10, but for the normalized wind speed bins in the Outer Range.**

The trial methods appear to have difficulties predicting power near rated wind speeds. The advantage of the trial methods in power prediction over the Baseline diminish for normalized wind speeds between 0.8 and 1 (Fig. 11). This nonlinear section of the power curve approaching rated power demonstrates weakness in power prediction within the current collection of trial methods. This feature amplifies after further outliers filtering (Fig. D2).

Den-Augturb is particularly skillful in power prediction over the Baseline method above rated wind speed in the Outer Range (Fig. 11). Even after removing outliers and reducing alpha, the individual improvement percentage for high winds stays in the 70s for Den-Augturb, unlike the considerable percentage reductions for other trial methods (Fig. D2a). Moreover, the average improvements via Den-Augturb remain statistically significant in three HWS bins (Fig. D2b). Den-Augturb also illustrates such leverage in the Outer Range WS-TI analysis by being the only trial method to reduce prediction uncertainty for both HWS-LTI and HWS-HTI bins (Fig. 10 and D1).

In some cases, outliers lead to notable prediction-error reductions of a trial method. The overlapping NME distributions suggest that the Den-Augturb method yields analogous power-prediction errors than the Baseline method near cut-in wind speeds (Fig. 12a). With the aid of the Den-Augturb correction, only 50% of the data sets improve from the Baseline (Fig. 11a and 12c). Above rated wind speed, the Den-Augturb method tends to correct for the Baseline's tendency to overpredict power (Fig. 12b). A few data sets report extreme improvements (Fig. 12d); thus, the distribution invalidates the Gaussian assumption of the t-test. Even after excluding those samples (to the left of the red dashed line in Fig. 12d), the Den-Augturb

adjustment at the above-rated wind speeds still significantly improves from the Baseline method (Fig. D2b). We recognize the limits of the t-test caused by the small sample size and the impacts of outliers; hence, we use bootstrapping to justify the t-test results.

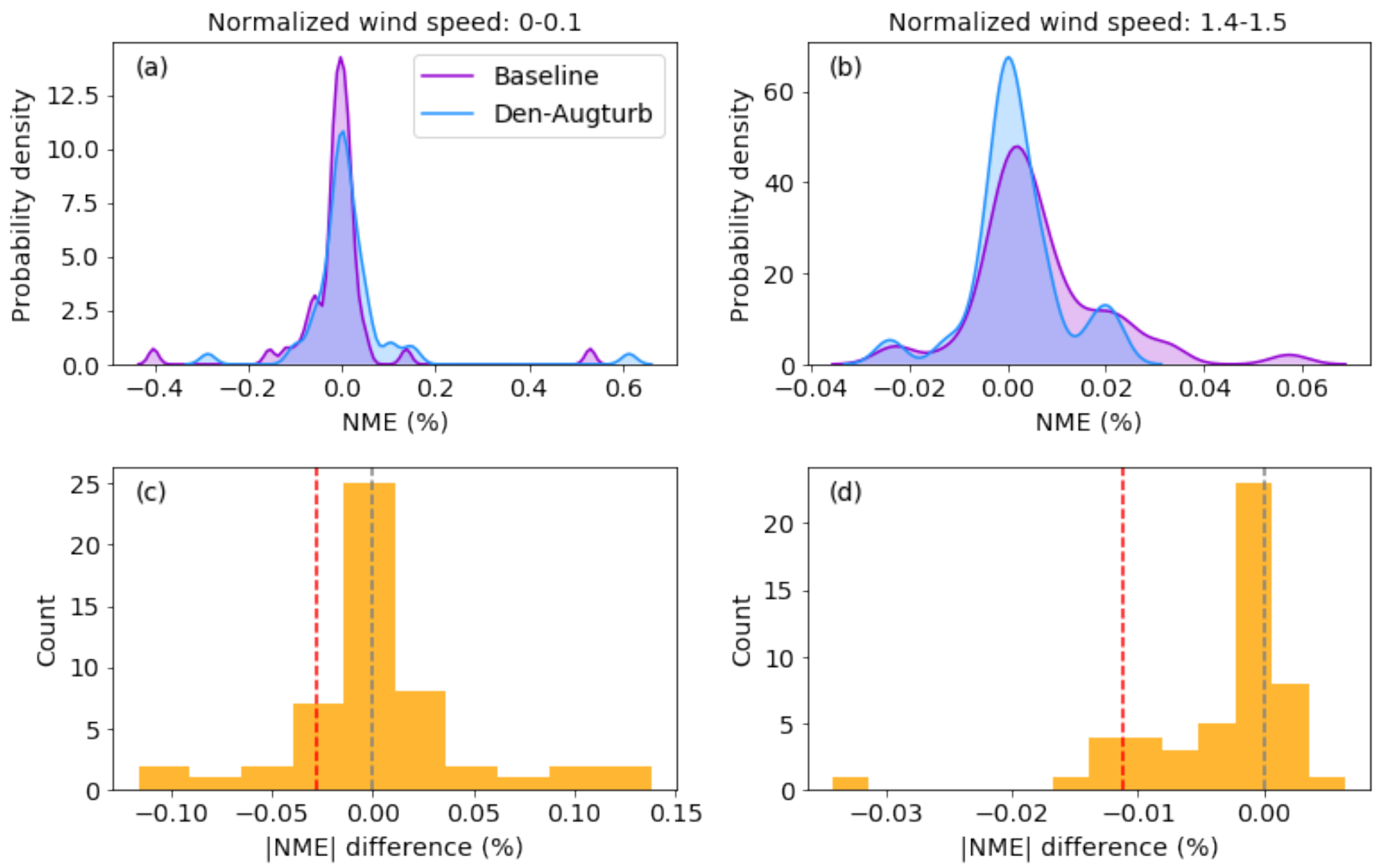

**Figure 12: (a) Probability density distribution of NME from the Baseline method (purple) and the Den-Augturb method (blue) for the Outer Range normalized wind speeds between 0 and 0.1; (b) as in (a), but for the Outer Range normalized wind speeds between 1.4 and 1.5; (c) histogram of differences of absolute NMEs between the Baseline and the Den-Augturb shown in (a); (d) as in (c), for the differences shown in (b). The grey and red dashed lines denote the zero NME difference and the 10$^{th}$ percentile of NME differences, respectively. Note that the ranges of the panel axes differ.**

### 4.3.3 Bootstrap analysis

Results from bootstrapping assert the findings from the statistical analyses in Sect. 4.3.2. We use bootstrapping to validate the statistical significance of improvement upon the Baseline method (Fig. 13). Thanks to the nature of this statistical technique, bootstrapping only provides guidance on the mean effect of the trial methods rather than the specific error reduction for a particular turbine. Therefore, for a large number of turbines, applying any of the trial methods significantly improves power prediction on average (Fig. 13a). Similarly, the Den-2DPDM and the Den-Augturb are respectively skillful for low-to-moderate and high wind speed scenarios for an average test case (Fig. 13b). Moreover, the coincidental bootstrapping findings reflect that the statistical test results (Fig. 10 and 11) are representative.

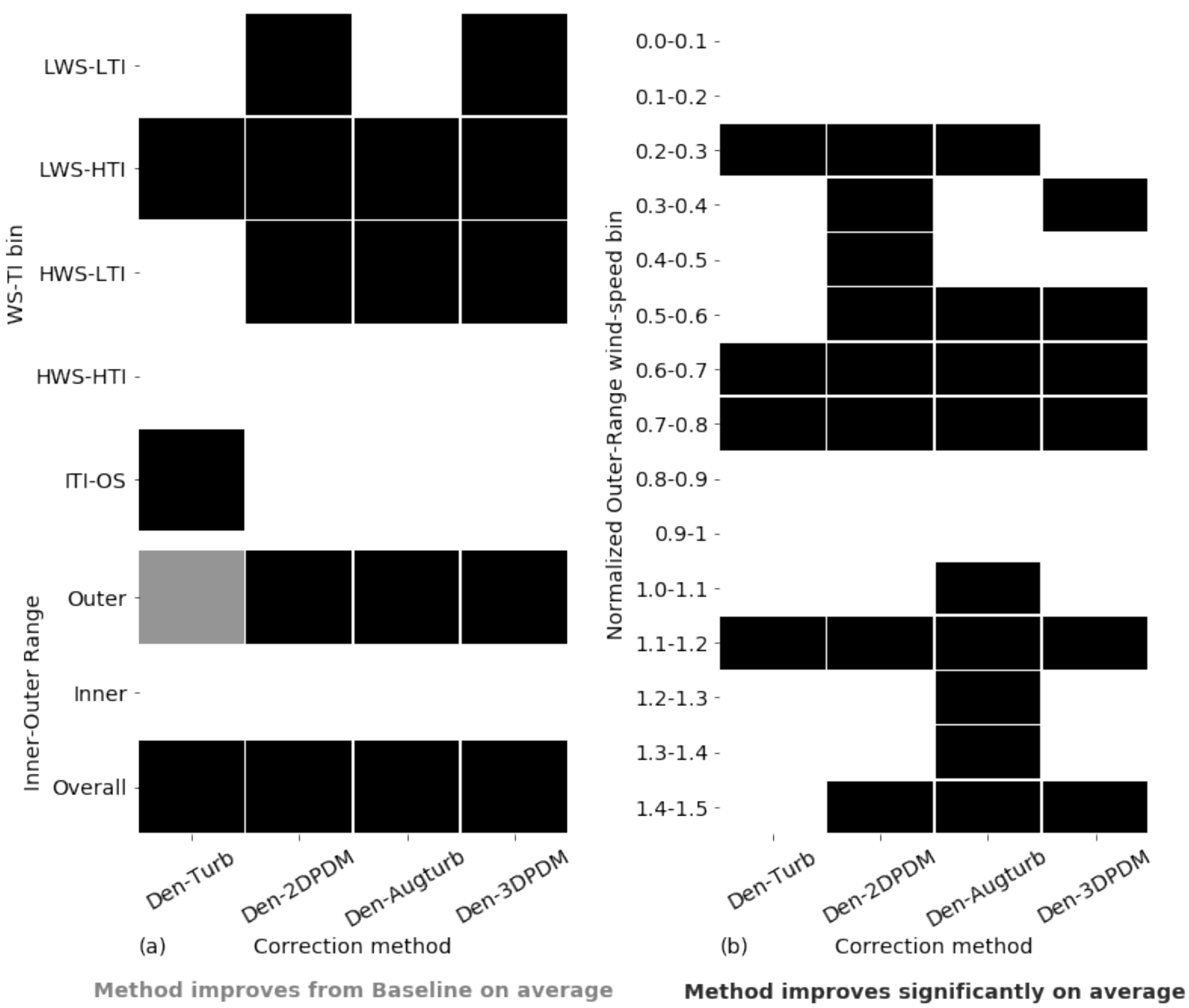

**Figure 13: Heat maps of the matched-pair t-test results using the means of the 10,000 bootstrapped samples. Passing the t-test indicates that the average of the 10,000 absolute NME means of a trial method is significantly smaller than that of the Baseline's. The heat maps are categorized into the four WS-TI bins and the ITI-OS bin in the Outer Range as well as the Inner and Outer Ranges (a), and normalized wind speed bins in the Outer Range (b). The bootstrapping only uses submissions without extreme improvements (Appendix C2). When a trial method demonstrates average improvement and statistically significant average improvement from the Baseline, the inflow condition is labeled in grey and black, respectively.**

We also perform the Levene's test on the 10,000 bootstrapped samples to evaluate the statistical significance of uncertainty reduction by a trial method (not shown). The bootstrapping analysis affirms the statistically insignificant uncertainty reductions by any trial method as in Fig. 10 and 11.

Additionally, we perform the bootstrap hypothesis test (simulating samples with means that fulfill the null hypothesis and deriving p-value empirically) as well as the Wilcoxon signed-rank test (a nonparametric test comparing the Baseline and a trial method). Those results match well with the bootstrapped t-test results (Fig. 13); therefore, the bootstrap analysis herein is reliable.

#### 4.3.4 Lessons learned

To improve power-prediction corrections, the industry should consider choosing a rigorous PDM based on a diverse collection of data sets that accounts for different atmospheric inflows. A contributing member of the PCWG derives the PDMs tested in the Share-3 exercise using 16 data sets (Sect. 1.1), which does not cover most turbine models, meteorological conditions, and terrains. The industry needs to expand the reference data sets to develop a comprehensive PDM. Altogether, fully eliminating power-prediction errors requires a more extensive search for an optimal method with a more reliable PDM.

Power production at high wind speeds in the Outer Range requires attention. We spotlight the higher wind speeds because those conditions contribute heavily to energy production (Sect. 4.2); nevertheless, the trial methods demonstrate unremarkable improvements upon the Baseline method in the HWS-HTI cases (Fig. 10). Using Den-Augturb correction displays skills in power prediction above rated wind speeds in the Outer Range (Fig. 11), yet choosing the method does not reduce prediction uncertainty significantly across wind speeds (Sect. 4.3.2). Overall, the trial methods are more accurate than the Baseline in predicting power at HWS; however, the corrections are imprecise.

Precise and comprehensive data sharing is the key to advance the industry's capability in wind turbine power prediction. The data and metadata the PCWG collected in the Share-3 exercise cannot answer some of the research questions we originally raised. For example, we cannot derive meaningful conclusions based on the geography or the time of day of the power measurements. Meanwhile, the characteristics of the data sets have a stronger influence on the value of a trial method than the choice of the method itself (Sect. 4.3.1). Therefore, ideally with higher-quality data, the PCWG should examine the influences to prediction errors from the metadata and the correction methods in the next intelligence-sharing exercise.

Additionally, the low data resolution casts limitations on the statistical analysis in this study. For instance, the PCWG analysis tool produces error statistics by summarizing the 10-minute data, so the collected data are already generalized, with temporal signals removed. The input data from meteorological towers may also be noisy and undermine the accuracy of the collected samples. The contributing members of the PCWG also run the analysis tool individually. Such decentralized procedures produce potential user errors; thus, this analysis requires filtering of erroneous samples (Appendix C1). Even though the Share-3 exercise, the collected data, and this analysis are embedded with uncertainty, this study synthesizes the multiyear effort of the PCWG in moving the industry forward, sheds light upon the ideal combinations of power-prediction methods, and thus aims to be a part of the tangible contribution to the IEC 61400-15 group.

## 5 Conclusions

The goal of the Power Curve Working Group (PCWG) is to advance the skills of the wind energy industry in modeling wind turbine power performance in complicated atmospheric conditions. This study discusses the findings from the Share-3 exercise, which is an intelligence-sharing initiative of the PCWG, its analysis tool for data collection, and its definitions on Inner Range and Outer Range conditions. In addition to the background information of the Share-3 exercise, this study

summarizes the analysis based on the 55 power performance tests with modern wind turbines from nine contributing organizations.

In this study, we examine the performance of four correction methods of power prediction, including Density and Turbulence (Den-Turb), Density and Two-Dimensional Power Deviation Matrix (Den-2DPDM), Density and Augmented Turbulence (Den-Augturb), and Density and Three-Dimensional Power Deviation Matrix (Den-3DPDM). We use the Baseline method (an interpolation to derive power curve) as the reference case, and we contrast the improvements in power prediction of four other trial methods against that reference. We compare the correction methods using the normalized mean error (NME), which describes the long-term average bias of power prediction to actual power production. We also use the matched-pair t-test and Levene's test to quantify whether a trial method reduces average error and uncertainty compared to the Baseline in a statistically significant way. We bootstrap the data to increase the representativeness of the statistical tests, and we strengthen the statistical inference by excluding the samples with substantial improvements and applying the Bonferroni correction.

We evaluate the trial methods primarily for the high wind speed (HWS) conditions in the Outer Range. A majority of the meteorological conditions are classified as the Outer Range, where the power production deviates from the reference power curve. This finding agrees with our expectation because we need sufficient amount of Outer Range data to validate the trial correction methods. Given that the HWS scenarios correspond to a larger contribution to turbine power production, the trial methods are more accurate at predicting power production than the Baseline at HWS, but the trial correction methods are as imprecise as the Baseline. For more than half of the submissions, the data sets have a larger influence on the prediction error than the choice of the trial methods, which indicates the need for high-quality metadata for further analysis.

This work serves as a foundation for the progress to come. Looking forward, the lessons learned through the Share-3 exercise suggest possible activities for the next phase of the PCWG's intelligence-sharing initiative. Specifically, new trial methods involving PDMs based on broad data sets, machine learning, and remote-sensing devices (RSDs) could be applied and tested. Corresponding to the growing popularity of RSDs, we should increase the volume of RSD-based data sets in future iterations of the PCWG intelligence-sharing initiative.

Additionally, because of the shape of the power curve, we find that among the Share-3 submissions, data with moderate wind speeds that are close to rated wind speeds largely contribute to the energy production. The existing wind-speed and turbulence-intensity (WS-TI) definitions, with only low and high bins, do not offer a proper arrangement for us to analyze such data comprehensively. Therefore, the next share exercise should consider further dividing wind speeds into bins of low, medium, high, and rated wind speeds. We should also consider the data with normalized wind speeds above 1.5, which heavily impact power production. Eventually, we use our findings to contribute toward the IEC 61400-12 and 61400-15 standards.

Data sharing shapes the future of the wind energy industry. Ultimately, sharing the 10-minute power performance data—although it requires a sea change of attitude across stakeholders—will fundamentally advance the wind industry in the most unimaginable ways. Despite the limited data we collected, this analysis demonstrates the importance as well as the implications of data sharing and should encourage future collaborations.

## Code availability

The PCWG analysis tool is hosted on GitHub at https://github.com/PCWG/PCWG.

## Data availability

The documents related to the Share-3 exercise are available at https://zenodo.org/communities/pcwg/. The example calculations of the zero-turbulence power curve and the corrected power at target wind speed and turbulence (Appendix A2) are also included in the repository.

## Appendix A

This section describes the correction methods tested in the Share-3 exercise. All of the methods discussed here use the Piecewise Cubic Hermite Interpolating Polynomial (Fritsch and Carlson, 1980) to derive the Inner Range power curve. Specifically, the interpolation recursively adjusts estimated power on the power curve to minimize prediction error in the Inner Range (Marmander, 2016). The PCWG analysis tool (Sect. 3.2) uses the "PchipInterpolator" in the SciPy package in Python (Jones et al., 2001).

The participants used the same Power Deviation Matrices (PDMs) in their Share-3 submissions, so we can fairly examine the effectiveness of the PDMs in correcting power predictions. A PDM expresses the expected power deviation between the observed data and the predictions using a specified Inner Range power curve. In Share-3, depending on the choice of the Inner Range definition (Sect. 3.1), the analysis tool automatically applies one of the three versions of the 2D and 3D PDMs for each data set. The PDMs are included as part of the source code of the PCWG analysis tool (version 0.8.0). We document the code and provide the repository in the Data Availability section.

Example calculations of the following correction methods are documented as Microsoft Excel files, and they are also included in the repository listed in the Data Availability section.

### A1 The Baseline method

The accuracy of each correction method in predicting the Outer Range data based on Inner Range measured power curve is assessed relative to a reference method. In the derivation of the Inner Range power curve and subsequent predictions of Outer Range data, a density correction is implemented to calculate the normalized wind speed ($V_n$). The measured 10-minute average wind speed ($V_{10-minute}$) has been corrected to correspond to a constant reference density ($\rho_0$) in accordance with the methodology of the 2005 edition of the IEC 61400-12-1 standard (International Electrotechnical Commission, 2005):

$$V_n = V_{10-minute}\left(\frac{\rho_{10-minute}}{\rho_0}\right)^{\frac{1}{3}} \tag{A1}$$

The method of calculation for the average density in each 10-minute period ($\rho_{10-minute}$) is dependent on the nature of the data set provided and the user configuration. It is either calculated from supplied temperature and pressure data or provided directly in the input time series data set (i.e., previously measured or calculated by participant institution). The reference density ($\rho_0$) used here is 1.225 kg $m^{-3}$ for all data sets.

### A2 The Density and Turbulence (Den-Turb) method

The Den-Turb method consists of applying the density correction of IEC 61400-12-1 2005's standard (described in Appendix A1) in addition to the turbulence normalization method described in Annex M of the 2017 edition of the IEC 61400-12-1 standard (International Electrotechnical Commission, 2017). The turbulence correction method accounts for the impact of wind-speed variations about the mean in each 10-minute period as well as the nonlinearity of the power curve. The turbulence correction is broadly divided into two parts: the generation of the zero-turbulence power curve and the correction of the reference power curve to a reference TI experienced at a site (Stuart, 2018).

We summarize the essential steps of the turbulence correction below. For simplicity, the power curve, turbulence intensity, wind speed, and turbine power coefficient are abbreviated as PC, TI, WS, and $c_p$ respectively:

1. Use a reference (Inner-Range) PC that is valid for a specific TI, and identify that TI as the reference TI
2. Calculate the initial zero-TI PC
   - 2.1. Use the reference PC to:
     - 2.1.1. Calculate the available power for the specific rotor geometry using the cubic relationship between WS and power; the resultant available power should always be larger than the reference power at each WS
     - 2.1.2. Identify the four reference-PC parameters: the cut-in WS, the rated power, the rated WS, and the maximum $c_p$
   - 2.2. Use the four reference-PC parameters as inputs to construct a PC for each WS:
     - 2.2.1. For WS below the input cut-in WS, assign zero power
     - 2.2.2. For WS above the input rated WS, assign the input rated power
     - 2.2.3. For other WS, preserve the cubic dependence of power on WS and use the input $c_p$ to calculate power. To account for the impact of TI on WS variation, each WS is expanded to a Gaussian distribution, where the standard deviation is the product of the WS and the reference TI. The resultant expected power at each WS is the sum of products between the zero-TI power and the WS distribution.
   - 2.3. If the resultant PC fulfills all three convergence criteria (when the cut-in WS, the maximum power, and the maximum $c_p$ converge to those of the reference PC):
     - 2.3.1. Label that PC as the initial zero-TI PC, and select the four input PC parameters (the cut-in WS, the rated power, the rated WS, and the maximum $c_p$) as the four initial zero-TI PC parameters
     - 2.3.2. Otherwise, adjust the four reference-PC parameters as revised inputs, repeat step 2.2 for a maximum of three times, or until the convergence criteria are met

3. Calculate the final zero-TI PC
   3.1. Use the four initial zero-TI PC parameters to construct a PC:
      3.1.1. For WS below the initial zero-TI cut-in WS, assign zero power
      3.1.2. For WS above the initial zero-TI rated WS, assign the initial zero-TI rated power
      3.1.3. For other WS, use the initial zero-TI $c_p$ and the available power to calculate power, and the resultant power would be valid for the specific TI
      3.1.4. Label the PC as the final zero-TI PC, and its maximum power can exceed that of the reference PC
4. Apply the final zero-TI PC to derive the turbulence correction
   4.1. Derive the simulated TI PC at the reference TI, where the power at each WS is the sum of product between the initial zero-TI power and the Gaussian WS distribution
   4.2. Finally, calculate the turbulence-corrected PC:
   corrected PC = reference PC + final zero-TI PC – simulated TI PC (A2)

### A3 The Density and Two-Dimensional Power Deviation Matrix (Den-2DPDM) method

The PDM correction method specifies a correction to be applied to power prediction for a given inflow bin of the data set. The PDMs used in the Den-2DPDM method define the correction to be applied dependent on normalized wind speed and turbulence binning. The correction in terms of wind speed and TI is the most common adoption of the PDM approach (Fig. 1 as an example).

As discussed earlier in Appendix A, the PDM applied to any given data set is dependent on the Inner Range definition used to derive the Inner Range reference power curve. The 2DPDM is applied based on the density-corrected wind speed as discussed in Appendix A1. The predicted power from the Inner Range power curve is thus corrected with a predetermined power deviation value for each specific normalized wind speed and TI.

One limitation of the 2DPDM is that the correction does not apply to the wind speed or TI bins with zero data counts (i.e., unpopulated bins). Depending on the site, measurements can be sparse at high wind speeds; hence, this correction becomes inapplicable for those inflow conditions.

### A4 The Density and Augmented Turbulence (Den-Augturb) method

The Den-Augturb method involves two steps: first the correction employs the Den-Turb method (Appendix A2), then the additional correction applies to the residual power deviation from the Den-Turb-corrected power curve. The method derives an empirical relationship between normalized wind speed and TI of the residual deviation, with the aid of a specific reference TI. For Share-3, the Den-Augturb method only applies to the normalized wind speeds below 0.9. The Den-Augturb method applies to the defined wind speed and TI bins regardless of the data counts in any particular meteorological conditions, which is an advantage over the Den-2DPDM method (Appendix A3). The calculation of the empirical turbulence is documented within the PCWG analysis tool, as listed in the Code Availability section.

For future iterations of the intelligence-sharing exercise, a possible modification to the current Den-Augturb method is to create a 2DPDM using the power deviation residuals and apply the PDM after the Den-Turb method (Appendix A2).

**A5 The Density and Three-Dimensional Power Deviation Matrix (Den-3DPDM) method**

The Den-3DPDM correction method is similar in nature to the Den-2DPDM method (Appendix A3). This correction method consists of three variables: normalized wind speed, TI, rotor wind speed ratio (Power Curve Working Group, 2016), which is defined as:

$$Rotor\ wind\ speed\ ratio = \frac{WS(Hub\ height + (\frac{3}{4} \times rotor\ radius))}{WS(Hub\ height - (\frac{3}{4} \times rotor\ radius))} \quad \text{(A3)}$$

where the rotor radius is half of the rotor diameter, and $WS$ denotes the wind speed at a given height.

We choose the rotor wind speed ratio over the shear exponent of the power law or the log law, because the magnitude of the shear exponent depends on the measurement heights. The same shear measured at two different height pairs yields two different shear exponents, where the shear exponent increases with decreasing hub height (Gollnick, 2015); whereas the rotor wind speed ratio accounts for the influence of hub heights and rotor diameters on wind shear over the rotor swept area and offers a fair and reliable depiction of shear across turbine models. Moreover, as per the Den-2DPDM correction, a 3DPDM is defined for each of the inner range definitions of Sect. 3.1.

**A6 Other methods**

We also implement other correction methods in the Share-3 exercise that require measurements at multiple heights, usually via remote sensing devices (RSDs). The shear normalization corrections in the form of the rotor equivalent wind speed (REWS) correction is applied to some of the participant data sets and reported to the independent aggregator. However, results pertaining to shear normalization corrections are not discussed in this study because the sample of those data sets is too small to draw statistically meaningful conclusions. Typically, an RSD is used to acquire data sets suitable for application of REWS and similar corrections; therefore, increased attention should be placed on increasing the volume of RSD-based data sets in future iterations of the PCWG intelligence-sharing initiative.

## Appendix B

**B1 Matched-pair t-test**

To better understand the statistical significance of the improvement for each trial method, we perform the matched-pair t-test (Montgomery and Runger, 2014). This is essentially the Student's t-test on the distribution of differences between the Baseline method and each trial method, in terms of their absolute NMEs.

We choose a one-sample, one-sided matched-pair t-test using alpha of 0.05. In statistical testing, alpha is a predetermined probability level of rejecting the null hypothesis ($H_0$) when the null hypothesis is true. The null hypothesis of

this test is that the mean of the absolute-NME difference distribution is larger than or equal to zero. In other words, the null hypothesis is that the trial method performs on par with, or worse than, the Baseline in terms of absolute NME. The alternative hypothesis ($H_A$) is that the mean difference of absolute NMEs between a trial method and the Baseline is less than zero, which indicates the trial method works better than the Baseline method. The null hypothesis and the alternative hypothesis are mathematically presented as follows:

$$H_0: \frac{1}{n}\sum_i^n (|NME_{Method}(submission_i)| - |NME_{Baseline}(submission_i)|) \geq 0 \quad \text{(B1)}$$

$$H_A: \frac{1}{n}\sum_i^n (|NME_{Method}(submission_i)| - |NME_{Baseline}(submission_i)|) < 0 \quad \text{(B2)}$$

To reject the null hypothesis of this one-sided test, the resultant t-statistic needs to be negative and the resultant p-value (probability to observe the t-statistic) divided by 2 must be less than alpha. When the null hypothesis of a certain atmospheric condition (for example, in the Outer Range) of a trial method is rejected, that means the improvement of such method upon the Baseline in the specific condition is statistically significant.

**B2 Levene's test**

We also perform the Levene's test (Brown and Forsythe, 1974; Gastwirth et al., 2009; Levene, 1960), which is a statistically robust version of the F-test, which compares the variances of two sample distributions. The objective of the Levene's test is to determine the statistical significance of the difference between two sample variances. An advantage of the Levene's test over a typical F-test is that the Levene's test works for nonGaussian distributions. We perform the Levene's test to a trial method only when the variance of that method's NMEs is smaller than the Baseline's.

In contrast to the matched-pair t-test on the differences of absolute NMEs, we apply the Levene's test on the NME distributions of the Baseline method and a trial method. We select an alpha of 0.05 for all the Levene's test. The null hypothesis is that the variance of the NMEs from the Baseline equals to the variance of the NMEs from a trial method, and the alternative hypothesis is that the two entities differ. The null hypothesis and the alternative hypothesis are mathematically presented as follows:

$$H_0: variance(NME_{Baseline}) = variance(NME_{Method}) \quad \text{(B3)}$$

$$H_A: variance(NME_{Baseline}) \neq variance(NME_{Method}) \quad \text{(B4)}$$

To reject the null hypothesis, the resultant p-value has to be smaller than the predetermined alpha. Because we only perform the Levene's test when a trial method's NME variance is smaller than the Baseline's, when the null hypothesis of the trial correction method for a certain atmospheric condition is rejected, the trial method reduces uncertainty in power prediction from the Baseline method with statistical significance. In general, few subsets of the submissions across atmospheric conditions pass the Levene's test, implying that the trial methods do not reduce uncertainty from the Baseline in power prediction in most cases.

### B3 Bootstrapping

To consolidate the statistical inference, we draw on the matched-pair t-test and the Levene's test using the limited size of the collected samples, we resample the submissions of the same sample size with replacement for 10,000 times, in which the process is also known as bootstrapping (Wilks, 2011). Bootstrapping preserves the same empirical distributions of the data, and each bootstrap sample matches the size of the observed sample.

For each bootstrap sample, we calculate the mean of the absolute-NME differences as well as the two variances of the Baseline method's NMEs and a trial method's NMEs. For each inflow bin, we perform the matched-pair t-test using the 10,000 bootstrapped means, which is approximately Gaussian according to Central Limit Theorem. In Fig. 13, for each bootstrap iteration, we select samples of the Baseline-trial-method NME pairs randomly from the data submissions. For each simulated subset of data, we calculate the mean absolute-NME difference, and we perform one t-test using the 10,000 means in Fig. 13. Furthermore, we also perform the Levene's test between the Baseline method and the trial method for each bootstrap sample. For each data bin, we calculate the fraction of the 10,000 bootstrapped samples that pass the Levene's test (not shown).

Fundamentally, the objective of bootstrapping is to assess the representativeness of the results from the matched-pair t-test and the Levene's test using the given collected set of submissions before any outlier removal. Note that we bootstrap using all the data after filtering out the erroneous samples (Appendix C1) as well as excluding the substantially improved data samples from the Baseline method (Appendix C2). Depending on the data bin, the postfiltering sample size varies between 41 and 46 data sets.

### B4 Kolmogorov-Smirnov (K-S) test

One limitation of the t-test (Appendix B1) is that it assumes Gaussian sample distribution. We perform the Kolmogorov-Smirnov (K-S) test (Wilks, 2011), which examines the goodness of fit between two distributions, on our samples. We use the K-S test with an alpha of 0.05. To validate the t-test, the null hypothesis of the K-S test we use states that the sample is drawn from the Gaussian distribution.

Because of the negative results from the K-S tests, the matched-pair t-test results have uncertainty. Meanwhile, based on the distribution shapes of absolute NME distributions (Fig. C2, for example), we consider our data samples as approximately Gaussian. After excluding samples with substantial improvements from the Baseline method (Appendix C2), nevertheless, the shapes of the distributions are closer to Gaussian qualitatively. Hence, we have strong confidence in the t-test results on filtered samples.

Although few samples across all the atmospheric conditions and trial methods pass the K-S test, real-world data are rarely perfectly Gaussian. Moreover, the K-S test is a highly stringent check for the Gaussian assumption. Therefore, the matched-pair t-test is still a useful tool in practice, and we implement various procedures, including bootstrapping (Appendix B3) and outlier filtering (Appendix C2) to make the t-tests as rigorous and valuable as possible.

eawe european academy of wind energy
Open Access WIND ENERGY SCIENCE DISCUSSIONS

## Appendix C

### C1 Filtering erroneous submissions

A key step for data quality control is to omit the submissions with the absolute Inner Range NMEs larger than 1%. Theoretically, each submission should record an Inner Range NME of zero. In other words, by definition the turbine should produce at or above capacity on average in the Inner Range. Hence, we exclude a total of three erroneous submissions with large, nonzero NMEs in the Inner Range (nonzero blue bars on the left in Fig. C1a). Note that all of the three submissions are from the same organization.

After filtering, the Inner Range NMEs hover around 0% (Fig. C1b); the Outer Range NMEs span almost 15% around 0% (Fig. C1c). In this manuscript, we only evaluate the 52 Inner Range NME-filtered submissions in Sect. 4, unless stated otherwise.

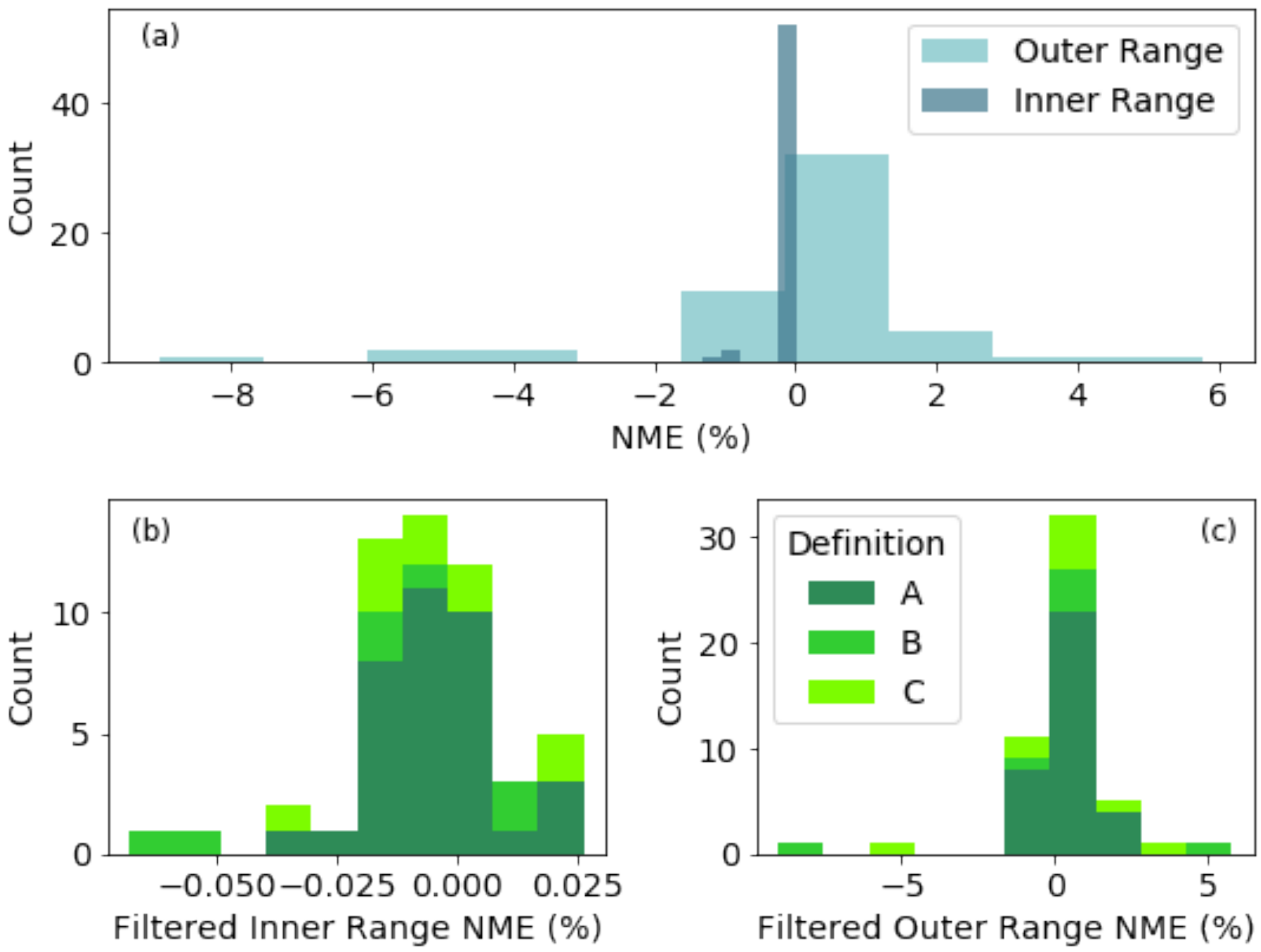

**Figure C1: Histogram of NMEs using the Baseline method: (a) NMEs of Inner Range (dark blue) and Outer Range (light blue) before filtering; (b) Inner Range NMEs after filtering out the submissions with absolute NMEs larger than 1%, categorized into three Inner Range definitions—definition A in dark green, definition B in green, and definition C in lime; (c) Outer Range NMEs categorized into three Inner Range definitions with the same color scheme as in (b).**

As stated in Sect. 3.4, we introduce a fifth bin of Inner Range TI and Outer Range wind shear (ITI-OS) for those Outer Range data not characterized by the four basic WS-TI bins. In four of the 52 NME-filtered submissions, some of the 10-minute Outer Range data are double counted in the four WS-TI bins and the ITI-OS bin, caused by the binning arrangement

of the PCWG analysis tool. Therefore, we only analyze the other 48 proper data sets for the WS-TI-related analysis in this study.

### C2 Filtering submissions with substantial improvement

For parts of the statistical analysis, we remove the few data sets when a trial method demonstrates substantial improvements upon the Baseline method. Specifically, we exclude the submissions in the 0.1 quantile, or the $10^{th}$ percentile of the absolute-NME differences between the Baseline and a trial method in each atmospheric condition. After this filter, the remaining samples are thus not skewed by remarkably improved submissions; in return, the statistical inference from the matched-pair t-test (Appendix B1) and the Levene's test (Appendix B2) becomes more rigorous. Note that this filter is only applied to the results in Sect. 4.3.3 and Appendix D.

For example, we filter out the most negative submissions in Fig. C2. The overall NME in each submission summarizes all 10-minute data points with a single value from each trial method. The absolute-NME difference of each submission between the Baseline and a trial method in Fig. C2 determines whether the trial method improves or worsens from the Baseline across atmospheric conditions. In other words, we illustrate the distributions of the contrasts between the Baseline and a trial method in Fig. C2. Across trial methods, the samples with strongly negative differences of absolute NMEs display the outliers of considerable improvements from the Baseline (Fig. C2). Those samples remarkably influence the average performance of a trial method, especially in the matched-pair t-test. As a result, removing them makes the t-test more rigid (Fig. D1 and D2). Note that the distributions of absolute-NME differences in Fig. C2, as well as the majority of the NME difference distributions we discuss in this study, do not pass the K-S test and are not strictly Gaussian.

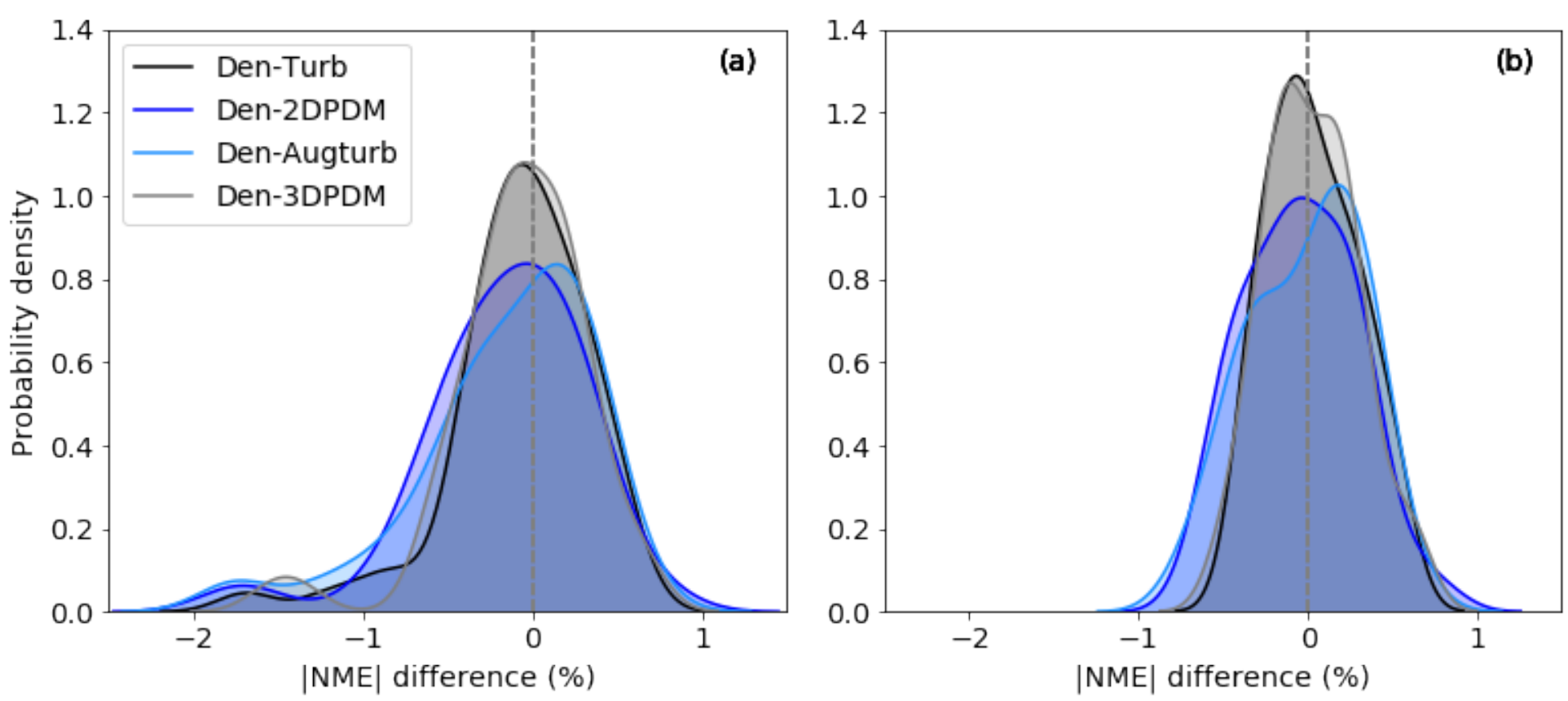

**Figure C2: (a) Probability density distribution of overall differences in absolute NMEs between the Baseline method and the four trial methods: Den-Turb in black, Den-2DPDM in blue, Den-Augturb in light blue, and Den-3DPDM in grey; (b) as in (a), but after filtering the 0.1 quantile of absolute-NME differences. Negative data points mean the specific trial method yields an overall absolute NME closer to zero than the Baseline for those submissions, and data with positive values represent such method results in a larger overall absolute NME than the Baseline. The grey dashed line in each panel marks the NME difference of zero.**

Of those samples with substantial improvements from the Baseline method across the four trial methods, a majority of those few submissions come from two organizations. Because of the limited metadata collected, we cannot draw any

eawe european academy of wind energy
Open Access WIND ENERGY SCIENCE DISCUSSIONS

meaningful conclusions on why those cases from the two organizations record considerable improvements upon the Baseline. After all, the input data set has a stronger influence on the degree of improvements from the Baseline than the choice of the trial method.

**C3 Bonferroni correction**

For the statistical tests we perform in this study, given an alpha of 0.05, each of the tests has a 5% chance leading to false positives, or a 5% chance of incorrectly rejecting the null hypothesis. Therefore, the more statistical tests we present simultaneously, the chance of yielding false positives becomes higher. This problem of multiple testing can be addressed by applying the Bonferroni correction to reduce alpha (Wilks, 2006, 2011), in which we divide alpha by the number of bins in each data category for each trial method. For example, we divide alpha of 0.05 by 5 for the four WS-TI bins and the ITI-OS bin for the WS-TI analysis in Fig. D1. We use a reduced alpha for every matched-pair t-test and Levene's test for each trial method. Note that this filter is only applied to the results in Appendix D.

Overall, the Bonferroni correction serves a precautionary purpose. Because we perform multiple statistical tests across bins of the inflow categories, we reduce the error rates of false positives for prudence. Each of the matched-pair t-test has its own null hypothesis, and the data samples are independent by nature. For example, in the top row of Fig. 10b, the null hypothesis is that a trial correction method does not yield smaller error from the Baseline in LWS-LTI condition in the Outer Range. In the second row of Fig. 10b, the null hypothesis is that the trial correction method does not yield smaller error compared to the Baseline in LWS-HTI cases. Both tests are independent, and both tests use distinct data sets, and specifically, data of LTI and HTI, by nature, do not overlap. Hence, a blanket reduction of alpha may make the tests overly rigorous. Nevertheless, the Bonferroni correction is useful so we present results with strict and trustworthy statistical inference (Fig. D1).

## Appendix D

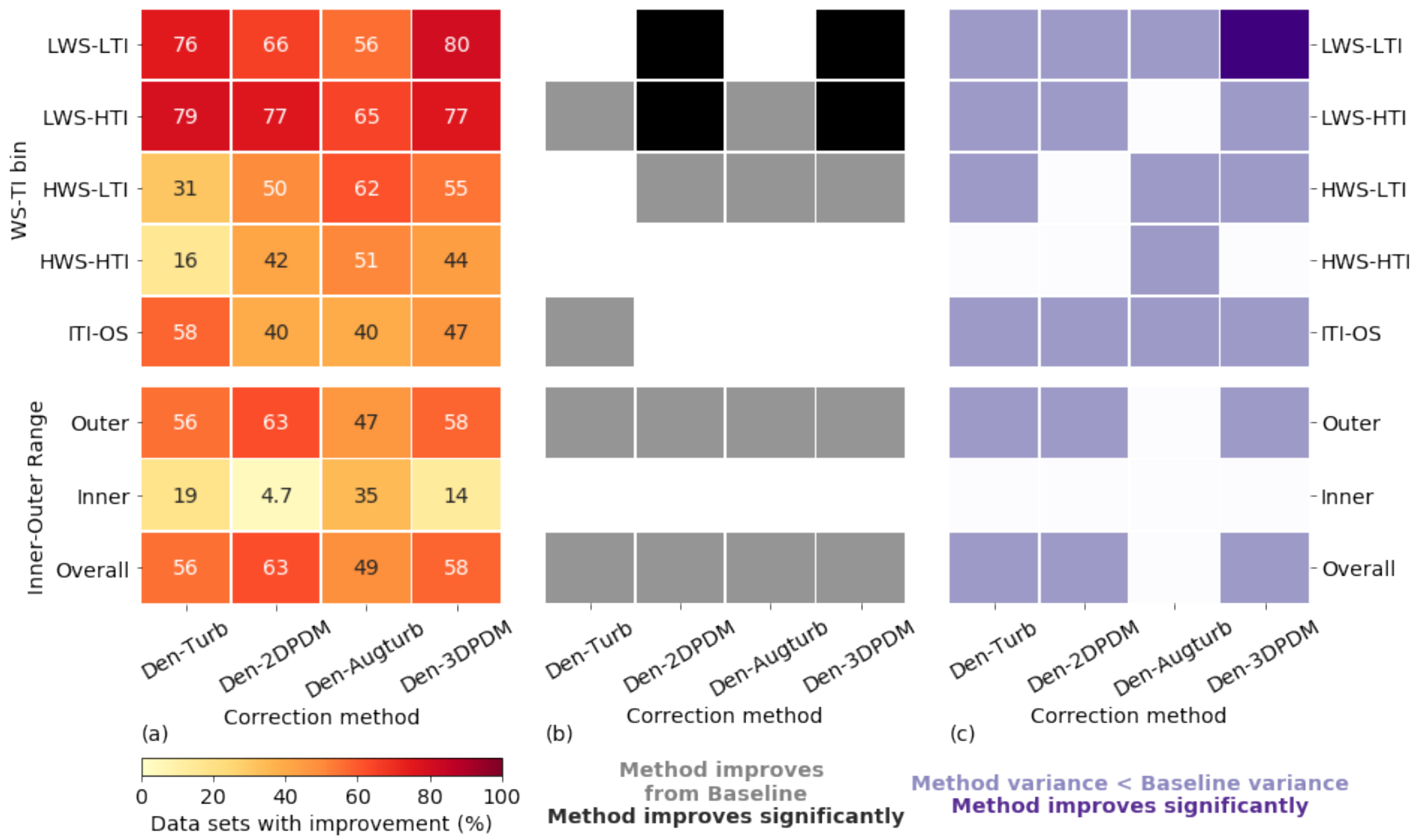

795 **Figure D1: As in Fig. 10, but after removing the submissions with a method's substantial improvement upon the Baseline method (top 10 percentile of the absolute-NME differences) as well as using smaller alphas based on the Bonferroni correction (the bin number is 2 for the Inner and Outer Ranges).**

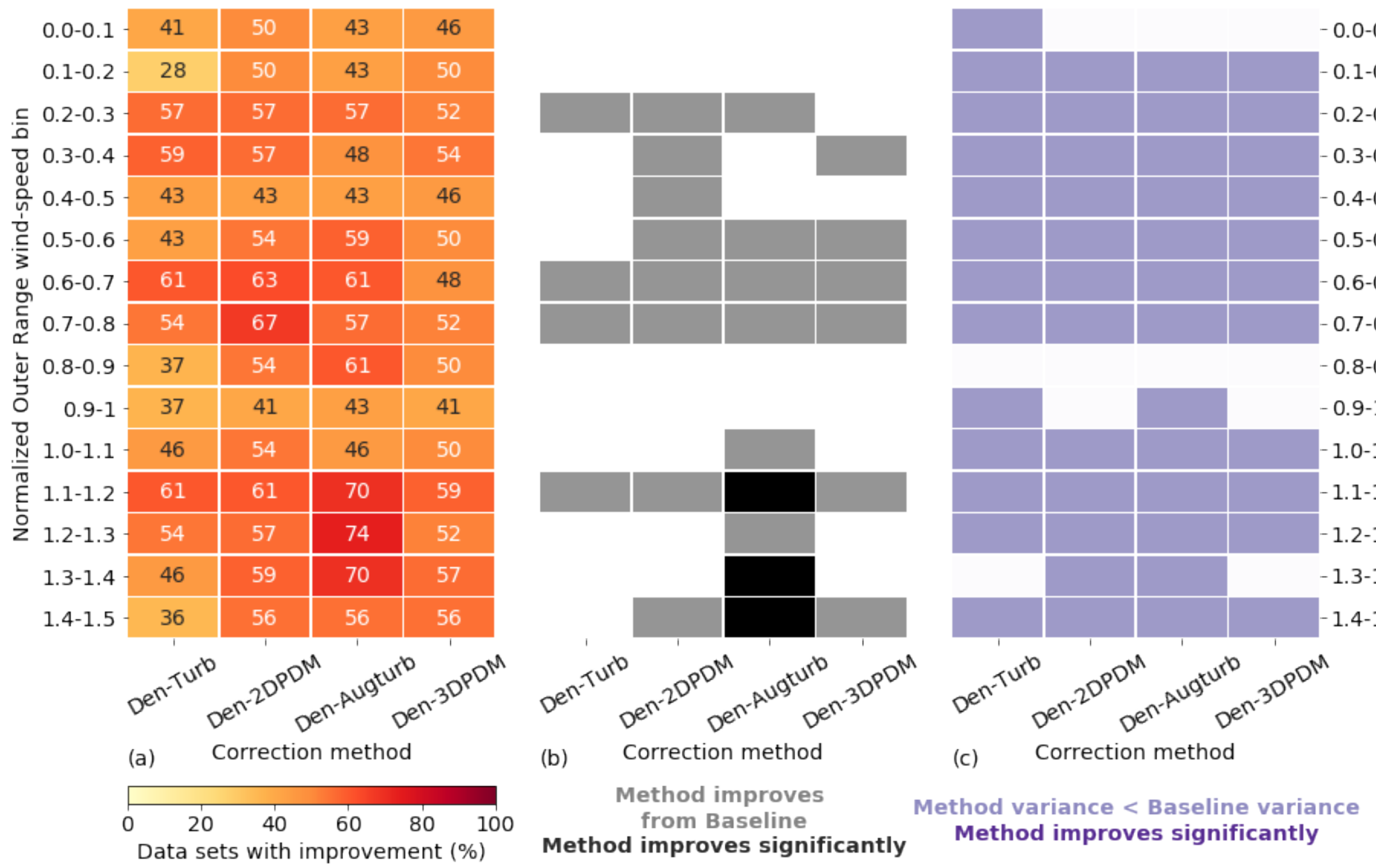

**Figure D2: As in Fig. D1, but for normalized wind speed bins in the Outer Range.**

## Author contribution

As veteran members on the PCWG organizing committee, AC, MJF, TG, PH, and PS formulated the research idea and designed the Share-3 exercise. LC, PH, and PS prepared the Python tool of the Share-3 data submission. MJF and JCYL facilitated the data collection at NREL. JPS and LW provided guidance on the statistical analysis. JCYL administered the data analysis, based on AC's previous work, and prepared the manuscript with the help of the coauthors. All coauthors edited and reviewed the manuscript.

## Competing interests

All of the authors indicate no conflict of interest.

## Acknowledgements

This work was authored by the National Renewable Energy Laboratory, operated by the Alliance for Sustainable Energy, LLC, for the U.S. Department of Energy (DOE), under Contract No. DE-AC36-08GO28308. Funding provided by

the U.S. DOE Office of Energy Efficiency and Renewable Energy's Wind Energy Technologies Office. The views expressed in the article do not necessarily represent the views of the DOE or U.S. Government. The U.S. Government retains and the publisher, by accepting the article for publication, acknowledges that the U.S. Government retains a nonexclusive, paid up, irrevocable, worldwide license to publish or reproduce the published form of this work, or allow others to do so, for U.S. Government purposes.

The authors would like to thank the members of the PCWG organizing committee. The authors express gratitude to the contributing members of the PCWG who submitted results to the Share-3 exercise: Barlovento, EDF, Innogy, LR, Mainstream, Renewable Energy Systems, Sevion, SSE plc, and Wood, where their submissions made this work possible. The authors would also like to thank our colleagues at NREL including Julian Quick, Tami Sandberg, and the NREL library staff.

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
