# Peer review of "The Power Curve Working Group's Assessment of Wind Turbine Power Performance Prediction Methods"

_Wind Energy Science, 2019_

## Referee Comment (RC1) · Ding Yu (Referee) · 11 Nov 2019

I am Dr. Yu Ding, Professor at Texas A&M University and the author of book "Data Science for Wind Energy." Despite being published only very recently, I am pleased to see that the authors cite this in their paper. It is also a great please to be asked to comment on this paper.

I personally participated in some of the PCWG meetings and had quite some discussions with Andy Clifton (when he was with NREL) and Jason Fields. I laud the collective effort of PCWG, especially the tremendous amount of work done initially by Peter Stuart and later by Taylor Geer for putting the group together, although I have unfortunately

not had the chance to meet either of them yet.

The work done by PCWG is important and will have a long-term impact for wind industry, particularly nowadays when government subsidies to wind energy are reducing or going away altogether. Performance enhancement and market competitiveness is the key to making wind energy self-sustainable. That is the common goal for all of us working in this area.

I am glad that the PCWG eventually summarizes this work for publication. When I was writing Chapter 5 of my book, I had been looking for a formal source to cite the work of PCWG. Had this paper been published earlier, I would have cited it in the book. Nonetheless, this is still not late and the paper will have good impact on wind researchers and practitioners alike. I do not have any serious concerns, barring some suggestions and comments below, and believe the paper should be published.

1. One thing rising to the level of a concern is the use of NME. I find it troublesome to use NME as the primary criterion to indicate fitting/prediction quality. Two functions, f and g, can be everywhere different, and the difference can be infinitely large, yet the NME can still be zero. This is different from NMAE, because when NMAE goes to zero, it guarantees the two functions, f and g, will converge to each other. In comparison studies or decision making, NME can be used but generally not as a primary criterion, and its use has to be carefully administered to serve a limited purpose, due to its fundamental theoretical flaw, namely when NME goes to zero, it does not guarantee anything concerning the similarity between f and g.

In the paper, it is said that the use of NME is for the long-term purpose while the use of NMAE is for short-term, and also that NME has a direct impact on P50. These statements are questionable. P50 means the 50-percentile (or the 0.5-quantile) of a probability distribution, which corresponds to the median of the probability distribution. To estimate a median, one minimizes the NMAE not NME. As a matter of fact, no statistical analysis or any objective function used in statistical modeling/machine learning/data science optimizes NME (because of NME's fundamental flaw). The statement saying that NME has a direct impact on P50 is not grounded in well-established scientific literature. This paper cited Clifton (2016) as the basis for this statement. Based on my understanding of Andy's technical rigor and prowess, I will be surprised if Andy supported the current statement (although he is a co-author of this paper).

To say that NME is for long-term while NMAE is for short-term is not principled either. In fact, NME can be particularly misleading when it is evaluated through a long-term average. Suppose that NME is calculated through a monthly average and its value is zero, but the day-ahead power prediction and the actual power can be different everyday, and the daily differences can be unboundedly large. NME being zero simply means that the prediction is more than the actual half of the times and less than the actual the other half of the times, but the more and the less cancelled each other. On surface, this zero NME indicates a perfect power curve model but an owner/operator using this model will lose money everyday (when doing day-ahead bidding). At the end of the month, the owner/operator suffers a huge monetary loss, and will puzzle how a "perfect" power curve model gets him/her into such a mess. On the other hand, if using NMAE, this problem won't happen.

As a side note, rigourously speaking, P50 is not the average (or mean) but the median. To estimate the mean (average), one should optimize an objective function based on RMSE (i.e., squared error loss). Of course, when a distribution is symmetric, like Gaussian, the two statistics, median and mean, become the same. Still, it is better to be rigourous when using the terminologies.

2. The paper identified PDM as an effective way to make accurate power prediction. This is not surprising. PDM is the same as multi-dimensional binning, which is a non-parametric method having sound data science foundation. The main limitation of PDM is its lack of scalability. One-dimensional PDM is not problem, 2D is fine, but 3D may be as far as PDM can go. If each variable is split into 20 bins, 3D PDM has 8,000 bins, and 4D PDM has 160,000 bins, and 5D PDM has over 3 million bins. But there are
only 50,000 or so 10-min data points for a whole year. In the end, the number of bins that need data points increases so fast in a multi-dimensional data space and they will far exceed the number of data points available, an issue known as the curse of dimensionality. This argument was made in our paper, Lee, Ding, Xie, Genton (2015) "Kernel Plus method for quantifying wind turbine upgrades," Wind Energy, 18: 1207-1219. It is the fundamental reason why I believe that data science/machine learning methods will in the end win out, because they can handle the scalability better (See Chapter 5, Data Science for Wind Energy). I hope that the authors are amendable to make a comment on the issue of "curse of dimensionality".

3. When PCWG started in 2012, the trend of data science methods and its practice was not yet popular or widely aware of outside the CS/Tech field. Back then, PCWG designed its specific way for intelligence sharing for understandable reasons. It is still a good practice. Nevertheless, after seven years, I hope that PCWG can adopt some of the tactics used by other academic/industry communities for identifying the best practice. For instance, the image processing challenge of 2013 administrated by CVPR identified the deep convolutional neural network (CNN) as the best tool for image processing (although people still try to understand, to this very day, why so). What the CVPR community did is to release a large amount of training image samples and reserve the test samples and ground truth for validation. The challenge administrator did not impose any restriction in terms of methods or programming languages. The testing results are done and averaged over thousands of test samples to effectively reduce biases.

If PCWG is interested, it can do something similar to CVPR's image processing challenge. Specifically, PCWG can goes through the following steps:

a. Collect enough datasets with wind/environmental inputs and power outputs from a wide variety of turbines. Normalize the power to [0, 1] and sanitize datasets to remove any farm/company identifying information.

b. Split the data into training and test. Release the training datasets, as well as the input data of the test sets. Both in pure text files.

c. A participating method, identified by a unique ID, known only to the participant and PCWG, makes prediction of power on the test set using the provided input and then produces a single column of power predictions.

d. PCWG's website engine takes the prediction and compute the error metric using the (never released) actual power output values. The prediction performance is released using the participant's ID. Again, I hope the error metric is not NME but a more principled choice.

4. The last comment of mine is about the different purposes that the IEC standard power curve method and a predictive power curve (used by owners/operators) serve. In my book, I have written the following, and wonder how PCWG in general, and the authors of this paper in particular, think about the difference.

Chapter 5, page 150, Data Science for Wind Energy, "Recall that we mention earlier in this chapter that the IEC method's intention is to provide a benchmark for verification purpose, rather than providing a method for real life performance assessment or wind power prediction. Consider the following analogy in the context of vehicle fuel economy. At the time of sale, a new car is displayed with a published fuel economy, in the unit of miles per gallon. The published fuel economy value is obtained under a standardized, ideal testing condition, which cannot be replicated in real-life driving. A car's real-life fuel economy based on someone's actual driving is almost always worse than the published one. In the wind power production, engineers observe something similar—using the IEC binning power curve often leads to a conclusion of under performance, which is to say that the actual power production falls short of the prediction.

Still, car buyers and car manufacturers feel that the fuel economy obtained under the ideal condition provides a reasonable benchmark, offering some ballpark ideas of how fast a car consumes its fuel. However, for consumers who care very much about the

real-life fuel economy, such as in commercial driving, they are not advised to use the published fuel economy value, as using the published value will surely lead to biased estimations. The same conclusion should have been extended to the IEC method, but in actuality, in the vacuum of robust, reliable, and capable power curve models, the IEC binning method is routinely used in the tasks or for the missions it is not designed for."

Once again, I commend the great work done by the authors and the remarkable effort in summarizing PCWG's intelligence sharing into a formal publication. I support its publication, although I hope the authors will take some of my comments into consideration while revising the paper (in particularly, NME). I look forward to continual interaction with the group and contributing to this important area.

---

## Referee Comment (RC2) · Anonymous Referee #2 · 20 Nov 2019

The paper studies the power curve modelling and deals with how its accuracy and uncertainty can be improved. The proposed work found to be interesting and the following comments can further improve the qualities of the paper contents. 1. Baseline method i.e, binning though being used by most turbine operators but has its own flaws such as data averaging issue, slow to respond, As discussed in below paper. The author should explore more about the weakness that baseline method had. Comparing the proposed technique with a technique that already have some issue, may affect the proposed model accuracy as compare to binning. 'Comparison of binned and Gaussian Process based wind turbine power curves for condition monitoring purposes' 2. Power is well known to be influenced by air density and this is reflected in the IEC Standard air
density correction procedure. IEC standard recommended air density correction does not give the most accurate power curve as suggested by the following articles. They have shown, power curve accuracy and uncertainty can be improved by adding air density correction instead of doing IEC precorrection. I think the paper must discuss this to improve the qualities of papers. 'Bulaevskaya V, Wharton S, Clifton A, Qualley G, Miller WO. Wind power curve modeling in simple and complex terrain using statistical models'. 'Incorporating air density into a Gaussian process wind turbine power curve model for improving fitting accuracy' 3. It would be great if the sample of data made public for a wide audience for improving the power performance of turbines.

Overall, the proposed work makes a significant contribution towards improving power curve accuracy as well as uncertainty.

---

## Author Comment (AC1) · 20 Nov 2019

We thank you for your support of this study and your recognition to the importance of this work. We also appreciate your past involvement with the Power Curve Working Group. Please see our response to your comments below.

1. The reviewer commented the use of NME in the manuscript. We understand that NME averages out positive and negative errors. The reviewer made a valid point that using NME does not reflect the error of power production from each 10-minute period. Meanwhile, the focus of this study is to evaluate the correction capabilities on the overall bias of each trial method over long periods, hence we choose to use NME instead

of NMAE. We want to investigate whether a correction method generally overpredicts or underpredicts production in different meteorological conditions, and NME is valid for such purpose. Moreover, because the metadata and the data sample size vary in the collected submissions, we primarily discuss the average bias in this study, as a foundation for future analyses.

NMAE is useful to evaluate the prediction error in specific, short-term events, while the metric does not provide information on the direction of bias. In this study, our focus is on average bias rather than the cumulative error of a correction method. In fact, the statistical results from NMAE are analogous to those of NME, please see the Fig. 1 below. As the reviewer suggested, we will assess NMAEs in future study, when we have large sample size and higher quality data.

We want to clarify we cited Clifton et al. (2016) for the P50 definition, rather than "average bias having a direct impact on P50", which is not in the reference. We are also changing the P50 definition to median AEP, thank you for pointing it out. You can find the changes we made to the manuscript below, from lines 212 to 226.

"A positive NME means the correction method overpredicts power production in over half of the data samples. Generally, NME represents the average bias on power production of the correction method. Such bias on power-curve modeling affects the long term P50, which is the median expected AEP over many years of production and is used to inform investment decisions. Meanwhile, NMAE denotes the average cumulative error of every 10-minute sample in a data bin, which is applicable for short-term power-production forecasting and time series analysis, making NMAE a stricter metric than NME. In NME, however, the positive and negative 10-minute errors cancel each other. Overall, the statistical results of NME (Sect. 4) are analogous to those of NMAE (not shown). For our purposes, we are interested in analyzing the long-term power prediction bias, and hence, we only discuss the NME for the rest of this manuscript; NMAE is introduced here because the metric is also generated by the PCWG analysis tool (Sect. 3.2). "

2. The reviewer mentioned the downside of increasing the dimensions of PDM, thank you for pointing it out. We added a paragraph in the manuscript, from lines 664 to 667:

"Note that increasing the number of data bins by switching from a 2DPDM to a 3DPDM spreads the data samples thinner, and smaller sample sizes in each bin could weaken the overall statistical confidence of the correction method (Lee et al., 2015). Therefore, methods such as the regression tree ensemble (Clifton et al., 2013) provide solutions for such dimension expansion problem. "

3. The Power Curve Working Group has been working on the Share-3 exercise, as presented in this manuscript, since mid-2017. Gathering support from a large industry group took an enormous amount of time and effort from the organizers and the participants, hence we gradually added trial correction methods over the previous data share initiatives (Table 1). In fact, a machine learning subgroup within the Power Curve Working Group has been dedicated to exploring data science applications for future sharing exercises.

This manuscript is about applying data science and statistics to improve the current practice of power curve modeling. The sharing exercises of the Power Curve Working Group do not require the sharing of raw data because of data privacy concerns (Sect. 3.2), so the release of training data sets proposed by the reviewer could be difficult to implement. In the last paragraph of the Conclusions, we advocate for data sharing. When most of the members of the Power Curve Working Group agree to publicly disseminate their own data sets, we will consider using the data science approach suggested by the reviewer.

4. Turbine manufacturers provide turbine power curves based on a controlled environment, while some manufacturers provide additional power generation information based on the real-world, complex meteorological conditions. The IEC 61400-12 standard highlights the procedures on deriving empirical power curves, also known as site-specific power curves, through power performance tests on the field with real-world inflow conditions. Because discrepancies sometimes exist between the reference power curves (often provided by the turbine manufacturers) and those seen in testing, the Power Curve Working Group aims to explore the underlying sources of errors. The analogy the reviewer provided differs from the typical usage of the IEC 61400-12 standard in the industry.

* * *
[Figure]

NMAE

**Normalized Outer Range wind-speed bin**

| | Den-Turb | Den-2DPDM | Den-Augturb | Den-3DPDM |
|---|---|---|---|---|
| 0.0-0.1 | 42 | 75 | 50 | 60 |
| 0.1-0.2 | 54 | 62 | 62 | 63 |
| 0.2-0.3 | 71 | 69 | 58 | 63 |
| 0.3-0.4 | 79 | 65 | 63 | 67 |
| 0.4-0.5 | 85 | 67 | 75 | 63 |
| 0.5-0.6 | 81 | 67 | 75 | 65 |
| 0.6-0.7 | 73 | 67 | 65 | 65 |
| 0.7-0.8 | 67 | 58 | 60 | 56 |
| 0.8-0.9 | 35 | 63 | 73 | 56 |
| 0.9-1 | 46 | 58 | 52 | 50 |
| 1.0-1.1 | 65 | 77 | 75 | 67 |
| 1.1-1.2 | 67 | 75 | 81 | 75 |
| 1.2-1.3 | 62 | 63 | 79 | 62 |
| 1.3-1.4 | 59 | 49 | 75 | 43 |
| 1.4-1.5 | 36 | 42 | 60 | 32 |

(a)  Correction method

(b)  Correction method

(c)  Correction method

0  20  40  60  80  100
Data sets with improvement (%)

Method improves
from Baseline
**Method improves significantly**

Method variance < Baseline variance
**Method improves significantly**

**Fig. 1.** As in Fig. 11 in the manuscript, but using NMAE. The features generally match those in Fig. 11.

**Supplement:**

[revised manuscript text omitted]

---

## Referee Comment (RC3) · Anonymous Referee #3 · 21 Nov 2019

The scope of this work is to assess the effectiveness of various trial methods to correct wind turbine power curves so that they more realistically represent real-world performance. A data gathering tool was developed for this work in order to anonymously aggregate contributor's data and test the trial methods. Rigorous data filtering and statistical assessment is also performed on the data.

Overall, the paper is well written. In general the authors use a high number of acronyms throughout the article, which may confuse the reader and should be avoided where possible. The definitions of "inner range" and "outer range" are clear, however the authors reference these definitions before a detailed explanation is provided. In addition,

to give more context and add background to the work, clarification of how the "inner range" power curve compares to the manufacturer's power curve, which is commonly known at all levels in industry, should be provided. In section 3.1 the authors propose three definitions of "inner range", without explaining why three definitions are given. In the conclusions, a more thorough discussion of how these methods can be improved in the future, to yield more statistically meaningful improvement should be included. In appendix A2, when describing the Den-Turb correction method, at point 2.2.3. the definition of a 0-TI power curve is ambiguous. In particular, the authors state that "each WS is expanded to a Gaussian distribution, where the standard deviation is the product of the WS and the reference TI", which is not clear since a 0-TI power curve is being calculated. In appendix A3, when discussing the Den-2DPDM trial correction methods, the authors state "One limitation of the 2DPDM is that the correction does not apply to the wind speed or TI bins with zero data counts (i.e., unpopulated bins)" without elaborating on the reasons this happens, or indicating if a correction of the method is possible. Based on the above, considering that the core of the article and of the work is solid and has been conducted rigorously, the reviewer recommends publication in the Wind Energy Science (WES) journal.

---

## Author Comment (AC2) · 27 Nov 2019

We thank the reviewer for providing valuable comments to improve our manuscript. In the following, the reviewer's comments are numbered, followed by our comments beginning with "Response:".

1. Baseline method i.e, binning though being used by most turbine operators but has its own flaws such as data averaging issue, slow to respond, As discussed in below paper. The author should explore more about the weakness that baseline method had. Comparing the proposed technique with a technique that already have some issue, may affect the proposed model accuracy as compare to binning.

[Figure]

Response: We now included the following, between lines 584 and 586, to describe the shortcomings of the binning method:

"The interpolation requires the separation of data into different discrete bins, and inevitably averages out the sample variations within a bin. The predefined bin width also determines the dependency of power on wind speed, which can introduce systematic error (Pandit and Infield, 2018a, 2018b)."

The "slow to respond" feature of binning (Pandit and Infield, 2018a), mentioned by the reviewer, does not apply to our context because the correction methods we propose aim to correct for long-term bias in power curve modeling rather than turbine condition monitoring and fault detection.

2. Power is well known to be influenced by air density and this is reflected in the IEC Standard air density correction procedure. IEC standard recommended air density correction does not give the most accurate power curve as suggested by the following articles. They have shown, power curve accuracy and uncertainty can be improved by adding air density correction instead of doing IEC precorrection. I think the paper must discuss this to improve the qualities of papers.

Response: To discuss the other preferred methods of density correction, we added the following from lines 608 to 613:

"Note that the air density correction in the IEC 61400-12-1 standard, although often used in practice, assumes the air density remains constant within the 10-minute period (Bulaevskaya et al., 2015). Such assumption oversimplifies real-world meteorological conditions, especially when the observed air density substantially differs from $\_0$ (Pandit et al., 2019). Therefore, Using air density as an independent input in statistical models such as Gaussian process, neural network, and random forest, can lead to smaller power-curve prediction errors than using the air-density-adjusted wind speed (Bulaevskaya et al., 2015; Pandit et al., 2019)."

3. It would be great if the sample of data made public for a wide audience for improving the power performance of turbines.

Response: The participating members of the Power Curve Working Group agreed to keep their raw data and the error statistics confidential, so unfortunately, we cannot disclose their data to the public.

References

Bulaevskaya, V., Wharton, S., Clifton, A., Qualley, G. and Miller, W. O.: Wind power curve modeling in complex terrain using statistical models, J. Renew. Sustain. Energy, 7(1), 013103, doi:10.1063/1.4904430, 2015.

Pandit, R. and Infield, D.: Comparative analysis of binning and support vector regression for wind turbine rotor speed based power curve use in condition monitoring, in 2018 53rd International Universities Power Engineering Conference (UPEC), pp. 1–6, IEEE., 2018a.

Pandit, R. K. and Infield, D.: Comparative analysis of binning and Gaussian Process based blade pitch angle curve of a wind turbine for the purpose of condition monitoring, J. Phys. Conf. Ser., 1102(1), 012037, doi:10.1088/1742-6596/1102/1/012037, 2018b.

Pandit, R. K., Infield, D. and Carroll, J.: Incorporating air density into a Gaussian process wind turbine power curve model for improving fitting accuracy, Wind Energy, 22(2), 302–315, doi:10.1002/we.2285, 2019.

---

## Short Comment (SC1) · 4 Dec 2019

[revised manuscript text omitted]

---

## Short Comment (SC2) · 5 Dec 2019

Hello Joseph,

Thanks for your answer. I do not have further comments for now, than the ones I made in the pdf file https://www.wind-energ-sci-discuss.net/wes-2019-69/wes-2019-69-SC1-supplement.pdf.

I also came to think of the data https://www.rave-offshore.de/en/leikline.html, and https://pod.ore.catapult.org.uk/ (7MW turbine)which may be available to you for research purposes.

[Figure]

All the best Rémi

---

## Editor Comment (EC1) · Alessandro Bianchini (Editor) · 5 Dec 2019

Dear authors, please reply to the short suggestions of Reviewer #3 at your earliest convenience.

---

## Author Comment (AC3) · 5 Dec 2019

We thank the reviewer for conducting a deliberate review to improve our manuscript. In the following, the reviewer's comments are numbered, followed by our comments beginning with "Response:".

1. In general the authors use a high number of acronyms throughout the article, which may confuse the reader and should be avoided where possible.

Response: This manuscript describes a large data science and data sharing project, and we define many parameters in the manuscript to give the full context about this

multi-year effort for the readers, and thus the content is dense and is filled with acronyms. To address the reviewer's concern, we added the full phrases before using the acronyms for the first time, especially in the Figure captions (Figure 1, 5, 6, and 12), the Conclusions, and the Appendices. Table 3 also summarizes the abbreviations of the trial methods.

2. The definitions of "inner range" and "outer range" are clear, however the authors reference these definitions before a detailed explanation is provided. In addition, to give more context and add background to the work, clarification of how the "inner range" power curve compares to the manufacturer's power curve, which is commonly known at all levels in industry, should be provided.

Response: The current structure of the manuscript begins with an introduction of the challenges about the wind turbine power curve (Sect. 1.1), which involves the concept of the Inner Range and the Outer Range. Then we discuss the two definitions extensively as part of the methodology that the Power Curve Working Group specifically chose to use in the Share-3 exercise (Sect. 3.1). We understand that introducing the ideas in the Introduction without the details raises questions, hence immediately after the first mention of Inner Range and Outer Range in the Introduction, in line 34, we wrote "The definitions are discussed in detail in Sect. 3.1."

We also added the following from lines 217 to 220 to differentiate the Inner Range power curve and the reference power curve (often provided by the turbine manufacturers):

"Note that the Inner Range Power Curve is only valid for a subset of TI and wind shear conditions (Table 2), which resembles the premise of a typical reference power curve provided by turbine manufacturers. The Inner Range power curve is derived from the observed data, which differs from a reference power curve. We also do not use any reference power curves in this analysis because we do not require the participants of the Share-3 exercise to share them."

3. In section 3.1 the authors propose three definitions of "inner range", without explaining why three definitions are given.

Response: Thank you for your comment. Lines 159 to 163 now read:

"We outline three Inner Range definitions in the Share-3 initiative because the PCWG analysis tool (Sect. 3.2) uses a specific definition to derive an Inner Range power curve for each data set. Depending on the data set, one of the three definitions is applied. For a data sample, the PCWG analysis tool first uses definition A as the default. If the resultant Inner Range data count under definition A is small (Power Curve Working Group, 2018), then the tool would switch to definition B. If the Inner Range data size is again small with definition B, then the tool would use definition C."

4. In the conclusions, a more thorough discussion of how these methods can be improved in the future, to yield more statistically meaningful improvement should be included.

Response: The last three paragraphs of the Conclusions cover various approaches for future work, including expanding the data sample size, applying data-driven techniques, and modifying the wind speed bin definition. According to the reviewer's comment, we modified the following in the Conclusions (lines 555 to 560):

"This work serves as a foundation for the progress to come. Looking forward, the lessons learned through the Share-3 exercise suggest possible activities for the next phase of the PCWG's intelligence-sharing initiative. Specifically, new trial methods involving more comprehensive PDMs based on broad data sets, machine learning, and data from remote-sensing devices (RSDs) could be applied and tested. Corresponding to the growing popularity of RSDs, we should increase the volume of RSD-based data sets and thus the statistical significance of the analysis in future iterations of the PCWG intelligence-sharing initiative."

Moreover, when we discuss the limitations of different methods in Appendix A1, A4,

A5, and A6, we also include proposed changes and additions to the trial methods in future iterations. In this manuscript, we want to focus on the results from this Share-3 exercise and spend a modest amount on potential future endeavors.

5. In appendix A2, when describing the Den-Turb correction method, at point 2.2.3. the definition of a 0-TI power curve is ambiguous. In particular, the authors state that "each WS is expanded to a Gaussian distribution, where the standard deviation is the product of the WS and the reference TI", which is not clear since a 0-TI power curve is being calculated.

Response: Thank you for raising a great point, and your comment increases the reproducibility of this work. Step 2, which describes the derivation of a zero-TI power curve, is now edited and expanded, from lines 626 to 653:

"2. Calculate the initial zero-TI PC

2.1 Use the reference PC to:

2.1.1 Calculate the available power for the specific rotor geometry using the cubic relationship between WS and power; the resultant available power should always be larger than the reference power at each WS

2.1.2 Identify the four reference-PC parameters: the cut-in WS, the rated power, the rated WS, and the maximum cp

2.2 Use the four reference-PC parameters as inputs to construct a zero-TI PC for each WS:

2.2.1 For WS below the input cut-in WS, assign zero power

2.2.2 For WS above the input rated WS, assign the input rated power

2.2.3 For other WS, preserve the cubic dependence of power on WS and use the input cp to calculate power. At each WS, the zero-TI power is the product of thee WS and the available power at the WS. To account for the impact of TI on WS variation, each

WS is expanded to a Gaussian distribution, where the standard deviation is the product of the WS and the reference TI. The resultant expected power at each WS is the sum of products between the zero-TI power and the WS distribution.

2.3 Determine the resultant PC parameters

2.3.1 For each WS, if the resultant expected power is larger than the 10% of the product of the rated power and the WS, then label the WS as cut-in WS

2.3.2 For each WS, divide the resultant expected power by the available power to calculate cp

2.3.3 Across WSs, select the minimum cut-in WS, the maximum power, and the maximum cp

2.4 If the resultant PC fulfills all three convergence criteria (when the cut-in WS, the maximum power, and the maximum cp converge to those of the reference PC):

2.4.1 Label that PC as the initial zero-TI PC, and select the four input PC parameters (the cut-in WS, the rated power, the rated WS, and the maximum cp) as the four initial zero-TI PC parameters

2.4.2 Otherwise, adjust the four reference-PC parameters as revised inputs, repeat steps 2.2 and 2.3 for a maximum of three times, or until the convergence criteria are met"

6. In appendix A3, when discussing the Den-2DPDM trial correction methods, the authors state "One limitation of the 2DPDM is that the correction does not apply to the wind speed or TI bins with zero data counts (i.e., unpopulated bins)" without elaborating on the reasons this happens, or indicating if a correction of the method is possible.

Response: This limitation of 2DPDM is caused by data availability. Lines 675 to 678 now read:

"One limitation of the 2DPDM is that the correction does not apply to the wind speed or

TI bins with zero data counts (i.e., unpopulated bins), and no correction would be made to the data in those bins. For instance, such drawback takes place when the wind-turbine locations used to derive the PDM rarely measure high wind speeds (Fig. 1 as an illustration). Hence, this correction becomes inapplicable for those inflow conditions."

Per another reviewer's comments, we added a brief discussion on this data availability-dimensionality problem of PDMs. Lines 705 to 708 now read:

"Note that increasing the number of data bins by switching from a 2DPDM to a 3DPDM spreads the data samples thinner, and smaller sample sizes in each bin could weaken the overall statistical confidence of the correction method (Lee et al., 2015). Therefore, methods such as the regression tree ensemble (Clifton et al., 2013) provide solutions for such dimension expansion problem."

References

Clifton, A., Kilcher, L., Lundquist, J. K. and Fleming, P.: Using machine learning to predict wind turbine power output, Environ. Res. Lett., 8(2), 024009, doi:10.1088/1748-9326/8/2/024009, 2013. Lee, G., Ding, Y., Xie, L. and Genton, M. G.: A kernel plus method for quantifying wind turbine performance upgrades, Wind Energy, 18(7), 1207–1219, doi:10.1002/we.1755, 2015. Power Curve Working Group: PCWG 3rd Intelligence Sharing Initiative Definition Document. [online] Available from: https://pcwg.org/PCWG-Share-03/PCWG-Share-03-Definition-Document.pdf, 2018.

---

## Author Comment (AC4) · 5 Dec 2019

Hi Rémi,

Thank you for your comments. We will certainly consider the datasets you listed in the next phase of our share exercise. Please kindly let us know if you have other questions about the paper.

Kind regards,

Joseph

---

## Author Comment (AC5) · 5 Dec 2019

Hi Alessandro,

Thank you for the reminder, our response to reviewer 3 has been posted.

Kind regards,

Joseph

---

## Editor Comment (EC2) · Alessandro Bianchini (Editor) · 9 Dec 2019

Dear authors, based on the outcomes of the discussion and on your responses, I am encouragin you to post a final comment/rebuttal and also to submit a revised paper, which takes into consideration all the suggestions of the reviewers. Best regards,

Alessandro Bianchini

---

## Author Comment (AC6) · 10 Dec 2019

We thank the reader for conducting a comprehensive review to improve our manuscript. Specifically, we thank the reader for providing comments on some of our word choices, and the edits we made according to your comments notably improve the manuscript. The reader also recommended many additions to enhance the context of the manuscript, and we adapted a lot of your suggestions we find relevant.

In the following, the reviewer's comments are labeled by the line number in the original manuscript submission, followed by our comments beginning with "Response:".

Line 29: Maybe consider adding, already here, that the power curve does not depend only on the wind speed: *hub height* wind speed, and valid for a given air density, a range of turbulence intensities at hub height and a range power-law shear exponents across the rotor plane.

Response: Lines 28 to 30 now read:

"Current industry practices involve predicting power output using a power curve, which defines power production as a function of hub-height wind speed. Besides the traditional understanding of a power curve, wind power production also depends on other meteorological variables including air density, turbulence, and wind shear."

Line 40: , for a given wind speed distribution.

Response: The phrase "site-specific", used earlier in the sentence, already implies the test is performed for a specific wind-speed distribution.

Line 40: It is a bit unclear if the objective of the test is to compare AEPs, or binned power values. I believe it is both in the IEC61400-12, yet eventually the main output of the analysis is an AEP number + an uncertainty. In some (most?) TSAs the metric that is used is the AEP.

Response: In this sentence, we focus on the binned power values of the test. We believe that the AEP is ultimately more important, hence in this study, we focus on using NME instead of NMAE (Sect. 3.4). Lines 40 to 42 now read:

"The wind energy industry performs power performance tests on wind turbines to test the site-specific power production of wind turbines by calculating the difference between the power predicted by the reference power curve (often provided by the turbine manufacturers) and actual power production at different wind speeds."

Line 41: It could be interesting fpr the reader to refer to the IEC61400-12.

Response: We are discussing the standard in the next Section (Sect. 1.2) comprehensively, and we want to focus on the challenge of the power curve herein.

Line 43: At this stage, it could have been nice for the reader to know than this range is defined by hub height TI and power-law shear exponent over the rotor.

Response: When we introduce the Inner Range and the Outer Range, we immediately add a sentence for the readers to refer to Sect. 3.1 in line 35: "The definitions are discussed in detail in Sect. 3.1.". We choose not to discuss the full definitions in the Introduction because we want to highlight the power curve challenge in Sect. 1.1, and we are dedicating a full subsection for the Inner Range and Outer Range definitions in the manuscript.

Lines 44, 46, 105: Inner Range ?; lines 135 to 136: this is a bit unclear: do you refer to the power curve measured during prototype testing ? what is the reference power curve - the inner range power curve ?

Response: Thank you for asking for clarification. The reference power curve is not the Inner Range power curve, as discussed in Sect. 3.1 and 3.2. As suggested by another reviewer, lines 214 to 217 now read:

"Note that the Inner Range Power Curve is only valid for a subset of TI and wind shear conditions (Table 2), which resembles the premise of a typical reference power curve provided by turbine manufacturers. The Inner Range power curve is derived from the observed data, which differs from a reference power curve. We also do not use any reference power curves in this analysis because we do not require the participants of the Share-3 exercise to share them."

When we first mention the reference power curve in the manuscript, we now introduce more details. Lines 40 to 42 now read:

"The wind energy industry performs power performance tests on wind turbines to test the site-specific power production of wind turbines by calculating the difference between the power predicted by the reference power curve (often provided by the turbine

manufacturers) and actual power production at different wind speeds."

Line 55: This is a bit unclear to me, can you provide the definition of "Power Deviation" using a formula ?

Response: Power deviation = Observed power – Reference power (or Predicted power), and this is added in line 56.

Line 60: You mean that they are not known ?

Response: We are implying that data sharing is limited within the industry, hence the data are isolated within their own organizations. We edited the following from lines 62 to 63:

"Additionally, the data that could be most useful for improving power-curve modeling are typically isolated within the industry, they are not shared between organizations, and their usage is stymied by intellectual property agreements."

Line 69: The reader may want to know whether these methods are normative or informative.

Response: Informative methods are implied here since we are referring to the IEC standard.

Line: 78: Maybe add a reference.

Response: We do not have a reference that specifically discusses the omission of the correction methods. Herein, we highlight the need of correction methods. Lines 80 to 81 now read:

"More importantly, given the inaccuracy of power curve models, not employing any corrections leads to increased scatter of production measurements of the power curve."

Line 79: hub height

Response: We added that to the text.

Line 80: This may imply that it wasn't known prior to 2017, or 2005, while it has been known for longer time, see for instance https://www.osti.gov/servlets/purl/6348447 (1990) or https://backend.orbit.dtu.dk/ws/portalfiles/portal/55566391/ris_m_2632.pdf (1987)

Response: Thank you. Per your suggestion, the word "since" is removed.

Line 81: Maybe add that this reference used model data (i.e. no measurements).

Response: Lines 84 to 85 now read:

"Clifton et al. (2013) demonstrated that simulated wind shear and TI impacted power performance with respect to the manufacturer's power curve in a clear and systematic way."

Line 85: It is unclear which of these references deal with power and loads, or just power, or just loads.

Response: We combined the references in one sentence, because herein we simply want to demonstrate that meteorological variables other than wind speed also affect wind energy production.

Line 88: What does "modern" mean in this context ?

Response: "Modern" means new data and techniques, such as remote sensing data and machine learning models, which are discussed in the following sentences.

Line 89: Maybe add a reference.

Response: This is the topic sentence of the paragraph, and the references are listed in subsequent sentences in the same paragraph.

Line 95: Maybe worth explaining what this means.

Response: This sentence summarizes the work from others, so specifying the variables may cause more confusion. It should be understood that the multidimensional

model includes other meteorological variables than wind speed, as discussed in this Sect. 1.2. We adjusted this sentence, and lines 101 to 103 now read:

"Recently, machine learning and neural networks that derive multidimensional power curve models involving many meteorological variables have grown in popularity."

Line 95: Do you mean these have been adopted by the industry ?

Response: Not necessarily, and we do not have the evidence to either support or deny a widespread adoption in the industry. Herein, we imply that these techniques have been popular topics for research.

Line 108: in the Outer Range.

Response: We do not limit our purpose to only Outer Range conditions, because the Outer Range changes with Inner Range conditions. In general, the Power Curve Working Group wants to benchmark the model effectiveness in all conditions, and mostly focuses on the Outer Range in this share initiative.

Line 122: It is not very clear what these data are, I assume these are datasets that fullfil the requirements of the IEC61400-12 for power performance testing. Also, it is not clear whether the datasets have been shared with a common, trusted third party, or if the data remained with the data owner and only the results were shared.

Response: These data sets were submitted for previous share initiatives from our participants, they should fulfill the IEC requirements, and they are owned by different organizations. Line 128 now reads:

"âIJŞ indicates method included in trial with at least 30 applicable summary statistics data sets submitted by the participants."

Line 134: can it be thought of differently ?

Response: "Thought of" is now changed to "interpreted".

Line 138: Isn't it contractual (i.e. a risk-mitigation measure for the manufacturer) ?

Response: It is possible. Herein, we are introducing Inner Range and Outer Range as a part of our research methodology, which is proposed by the Power Curve Working Group.

Line 141: You mean, by averaging a number of outerange AEP results from different test locations, or at the same site using different outer range datasets ? What I focused only a a subset of the outer range leading to larger AEP ? This statement is a bit unclear.

Response: For a wind turbine, its resultant AEP would fall below its capacity when one accounts for all the Outer Range conditions it experiences. This is a key distinction between Inner Range and Outer Range, which is explained in this paragraph. It is possible that if one only focuses on a subset of Outer Range conditions, the "AEP" could theoretically be larger than that predicted by a reference power curve.

Line 145: power-law wind shear exponent

Response: Lines 156 to 158 now read:

"The PCWG differentiates Inner Range and Outer Range data based on the wind shear and TI. Wind shear, represented by the power law exponent, is calculated using the wind speeds between the lower blade tip and hub height, and the TI at hub height (Power Curve Working Group, 2018)."

Line 151: maybe worth explaining what this means in this context

Response: To be more specific, "under ideal conditions" is now changed to "in a controlled environment defined in the IEC standard".

Line 152: You may want to provide some explanation about the limitation of using a power-law shear measured between blade top and bottom tips. Maybe could you refer to some ABL meteorology works in flat terrain, like the ones of Alfredo Peña ? In particular, it could be interesting for the reader to understand how the wind speed profiles is characterised in the surface layer (MOST) and above (not only MOST), how the height of the surface layer changes with stability and therefore how the power-law may or may not represent well the wind profile. See examples in slides 12 and 13 of [C2W19]: deriving a power-law shear value computed only using the highest and lowest tip heights wind speeds would not represent accurately the wind speed profile over the rotor. The errors are, in absolute terms larger in stable conditions, but in relative terms the errors are also large in unstable conditions. This is, important to mention when dealing with tests that have been carried out using different rotor spans. Also, it is important to explain the link between wind shear and turbulence intensity, that is: the influence of the roughness and orography (mechanical turbulence) and the stabiity (thermal turbulence), so the reader know these are linked, but that this link is site-specific. For instance, large turbulence may still exist in stable conditions, because of an obstacle upstream, so the link is not a simple as "stable -> small TI/large shear". [C2W19]: http://c2wind.com/f/content/windeurope_wra_workshop_20190627_c2w_rev4.pdf

Response: Thank you for your comment. Similarly, one could argue our use of TI, or even AEP, omits their own limitations. We understand that representing the wind shear with the shear exponent can be inadequate in different ways, while discussing its imperfection herein is marginally relevant. This is a study about data sharing and data science on power curve modeling, rather than a discussion on meteorological variables and calculations. In this section, we emphasize on differentiating the Inner Range and the Outer Range, we explain our analysis methodologies extensively in Sect. 3 and Appendix C, and we also discuss the constraints of our methodology throughout the manuscript, especially in Appendix A and B.

Line 166: Are meta-data shared as well (rotor span for instance) ?

Response: Some of the submissions include their metadata including the rotor span, yet not all of them do.

Line 213: median ?

Response: That is correct, as pointed out by another reviewer. Lines 237 to 239 now read:

"Generally, NME represents the average bias on power production of the correction method. Such bias on power-curve modeling affects the long term P50, which is the median expected AEP over many years of production and is used to inform investment decisions."

Line 218: subsetting the overall dataset of 10-minute samples? "slicing the 10-minute data" may lead to thinking you are using sub-10minute time series.

Response: Thank you, and the phrase now reads: "slicing all the 10-minute data of each submission in several ways"

Line 231 to 239: Do you provide statistics about how much data is found in each category, for instance: for more than 80% of the tests, less than 30% of the data are in the inner range... or something like this, which could help visualise the populations ?

Response: Sect. 4.1, including Fig. 4 and Fig. 5, is dedicated to discussing the metadata distributions.

Line 240: What does it mean in this context ?

Response: The word "broadly" is unnecessary and thus removed.

Line 283: Could you also provide explanations about the wind measurements ? How did you derive the power-law shear across the rotor is only met mast at hub heights are used ? Same question for TI (lidar TI?) ?

Response: The participants do not only use wind speeds at hub height in the PCWG analysis tool, where wind shear "is calculated using the wind speeds between the lower blade tip and hub height, and the TI at hub height (Power Curve Working Group, 2018)." (lines 157 to 158).

Line 288: What does it mean in this context ?

Response: The word "modern" here means that the data we have indicate wind turbines of modern models, because most of the tests results are recorded after 2013.

Line 291: There are different meteorogical conditions and sites in each of these countries, maybe could the users have reported "regions" instead of countries ?

Response: Unfortunately, only a subset of the submissions indicate their region or country of origin, so we cannot fully disclose the geographical metadata.

Line 316: in the control region I (optimal TSR)

Response: Lines 351 to 352 now read:

"This feature fits our expectation because of the cubic relationship between wind speed and power, when the hub-height wind speed is between cut-in wind speed and rated wind speed."

Line 331: Are these the ones discussed in Section 4.1 ?

Response: Yes, the phrase is updated to "only 48 of the 55 submissions..."

Line 346: can you explain where these results are in Figure 8a (row, column) ?

Response: This is explaining the "ITI-OS", which is Inner Range TI and Outer Range shear, and it is the 5th row from the top.

Line 353: I read this paper quickly, sorry for the silly question, but: shouldn't this value be zero ?

Response: Thank you for raising this question. You are correct, and these numbers are very close to zero, which are less than or close to 0.01%. These numbers are not absolute zeros because the trial methods and the interpolation method (Appendix A) minimize the prediction error in the Inner Range, while the residual errors are not necessarily equal to zeros. Thanks to your comment, we added lines 387 to 388:

"Ideally, the Inner-Range errors would be zero, yet the trial methods and the interpolation method (Appendix A) minimize the prediction errors and do not necessarily result in zero residual errors in the Inner Range (second-last row in Fig. 8)."

Line 368: What does this mean, in this context ?

Response: The word "broadly" is unnecessary and hence removed. The key message here is that for some submissions, the data set itself influences how effective a trial method is.

Line 386: What does this mean in this context ?

Response: For the submissions we have on hand, we do not find the meaningful correlations between turbine characteristics and trial method effectiveness. However, we do not have all the metadata for all the data sets, hence we can only conclude this in a general sense.

Line 387: Still, i think it could be interesting to show the correlations plots.

Response: We decide not to display in this manuscript because of the lack of strong correlations. Mentioning this finding in the text sufficiently serves the purpose.

Line 456: is it probability density (unitless) ?

Response: The probability density here represents the probability per file count. The units of probability density is usually implied. Per your comment, we added the explanation in the caption for Fig. 12 and Fig. C2.

Lines 485 to 486: What does this mean, specifically ?

Response: It means the current set of power deviation matrices is not derived from a broad selection of data sets. This is the topic sentence of the paragraph, and the explanation is provided in the subsequent sentences.

Lines 505 to 506: Key message.

Response: Hence we put it in this section.

References

Clifton, A., Kilcher, L., Lundquist, J. K. and Fleming, P.: Using machine learning to predict wind turbine power output, Environ. Res. Lett., 8(2), 024009, doi:10.1088/1748-9326/8/2/024009, 2013. Power Curve Working Group: PCWG 3rd Intelligence Sharing Initiative Definition Document. [online] Available from: https://pcwg.org/PCWG-Share-03/PCWG-Share-03-Definition-Document.pdf, 2018.